# TMEM135 links peroxisomes to the regulation of brown fat mitochondrial fission and energy homeostasis

Donghua Hu[1], Min Tan[1], Dongliang Lu[1], Brian Kleiboeker [1], Xuejing Liu[1], Hongsuk Park[1], Alexxai V. Kravitz [2], Kooresh I. Shoghi[3], Yu-Hua Tseng [4], Babak Razani [5,6], Akihiro Ikeda [7] & Irfan J. Lodhi [1] ✉

Mitochondrial morphology, which is controlled by mitochondrial fission and fusion, is an important regulator of the thermogenic capacity of brown adipocytes. Adipose-specific peroxisome deficiency impairs thermogenesis by inhibiting cold-induced mitochondrial fission due to decreased mitochondrial membrane content of the peroxisome-derived lipids called plasmalogens. Here, we identify TMEM135 as a critical mediator of the peroxisomal regulation of mitochondrial fission and thermogenesis. Adipose-specific TMEM135 knockout in mice blocks mitochondrial fission, impairs thermogenesis, and increases diet-induced obesity and insulin resistance. Conversely, TMEM135 overexpression promotes mitochondrial division, counteracts obesity and insulin resistance, and rescues thermogenesis in peroxisome-deficient mice. Mechanistically, thermogenic stimuli promote association between peroxisomes and mitochondria and plasmalogen-dependent localization of TMEM135 in mitochondria, where it mediates PKA-dependent phosphorylation and mitochondrial retention of the fission factor Drp1. Together, these results reveal a previously unrecognized inter-organelle communication regulating mitochondrial fission and energy homeostasis and identify TMEM135 as a potential target for therapeutic activation of BAT.

The global prevalence of obesity continues to rise with the concomitant increase in the prevalence of type 2 diabetes resulting from the obesity-associated disruption of insulin action and β-cell function[1,2]. Thermogenesis mediated by brown adipose tissue (BAT) influences systemic metabolism by promoting resting energy expenditure, whole-body glucose disposal, and insulin sensitivity[3]. Thus, enhancing BAT activity represents a promising alternative strategy to treat obesity and the associated diabetes. Brown adipocytes and the related beige adipocytes are enriched in mitochondria and mediate thermogenesis through a mechanism involving uncoupling protein-1 (UCP1), a mitochondrial protein that disassociates respiration from ATP synthesis to generate heat, as well as other mechanisms that are independent of UCP1[4–6].

Increasing evidence suggests that mitochondrial morphology is also a critical regulator of the thermogenic capacity of an adipocyte, with brown adipocyte mitochondria being more fragmented and

[1]Division of Endocrinology, Metabolism & Lipid Research, Department of Medicine, Washington University School of Medicine, St. Louis, MO, USA. [2]Department of Psychiatry, Washington University School of Medicine, St. Louis, MO, USA. [3]Mallinckrodt Institute of Radiology, Washington University School of Medicine, St. Louis, MO, USA. [4]Section on Integrative Physiology and Metabolism, Research Division, Joslin Diabetes Center, Harvard Medical School, Boston, MA, USA. [5]Cardiovascular Division, Department of Medicine, Washington University School of Medicine, St. Louis, MO, USA. [6]University of Pittsburgh School of Medicine and UPMC, Pittsburgh, PA, USA. [7]Department of Medical Genetics, University of Wisconsin-Madison, Madison, WI, USA. ✉e-mail: ilodhi@wustl.edu

circular (particularly in response to β-adrenergic receptor stimulation) and white adipocyte mitochondria being more tubular in shape[7,8]. Mitochondrial morphology is controlled by mitochondrial dynamics. Mitochondria cycle between fused and fragmented states depending on the metabolic needs of the cell[9]. In certain cell types, such as cardiomyocytes, sustained mitochondrial fragmentation is associated with bioenergetics stress that may lead to lipotoxicity[10]. However, mitochondrial fission in brown adipocytes is not deleterious per se, since fragmented mitochondria exhibit increased uncoupled respiration, a *sine qua non* of thermogenic adipocytes. Fragmented mitochondria in thermogenic adipocytes are thought to promote uncoupled respiration activity by directing β-oxidation of fatty acids toward UCP1-mediated heat production instead of ATP synthesis[8,11,12].

Mitochondrial dynamics is regulated by specialized proteins that belong to the dynamin-like family of GTPases, including dynamin-related protein 1 (Drp1), mitofusins 1 and 2 (Mfn1/2), and optic atrophy-1 (Opa1)[13]. Drp1 is a cytosolic GTPase required for mitochondrial fission[14]. Mfn1/2 are involved in fusion of the outer mitochondrial membrane[15]. Opa1 is required for fusion of the inner membrane. Proteolytic processing of Opa1 is associated with inhibition of fusion and increased inner membrane fission[16]. Recent studies demonstrate that mitochondrial fission involves recruitment of multiple organelles, including the endoplasmic reticulum (ER), lysosomes and Golgi apparatus to mitochondria[17].

Peroxisomes, single membrane-bound organelles involved in lipid metabolism, have emerged as important regulators of energy homeostasis[18,19]. Peroxisomal biogenesis increases in brown and beige adipocytes in response to cold treatment in a manner dependent on the thermogenic co-regulatory protein PRDM16, suggesting that gene expression of factors involved in thermogenesis and de novo peroxisome formation is coordinated[20]. Adipose-specific knockout of the critical peroxisomal biogenesis factor Pex16 (*Pex16-AKO*), which results in absence of peroxisomes in adipose depots, impairs thermogenesis by inhibiting cold-induced mitochondrial fission due to decreased mitochondrial membrane content of the peroxisome-derived lipids called plasmalogens[20]. To delve deeper into the molecular mechanism underlying the role of peroxisomes in mitochondrial dynamics and thermogenesis, in the current study we performed untargeted proteome analysis on mitochondria isolated from BAT of *Pex16-AKO* and control mice. Our results revealed that the protein most decreased in mitochondria from the knockout mice is a transmembrane protein called TMEM135 (also called PMP52), which was originally identified by mass spectrometry as a protein enriched in peroxisomes isolated from liver and kidney[21,22]. Notably, TMEM135 has also been shown to be localized in mitochondria and promote mitochondrial fission[23], but the underlying molecular mechanisms have remained unknown.

Here we report that TMEM135 is a critical mediator of the peroxisomal regulation of mitochondrial fission in brown adipocytes. We show that acute β-adrenergic receptor activation promotes association between peroxisomes and mitochondria and peroxisomal lipid-dependent localization of TMEM135 in mitochondria, revealing a previously unrecognized inter-organelle communication regulating mitochondrial fission in brown fat. Our studies identify TMEM135 as an important regulator whole-body energy metabolism through its role in brown adipocyte mitochondrial fission. Suggesting translational relevance to metabolic homeostasis in humans, *Tmem135* gene expression is decreased in subcutaneous white adipose tissue (WAT) of individuals with obesity. Moreover, we describe a BMI-associated coding variant of *Tmem135* that impairs mitochondrial fission in human brown adipocytes. These studies suggest that TMEM135 might be an attractive target for therapeutic activation of thermogenic fat, perhaps leading to a treatment option for obesity and its associated metabolic disorders.

## Results

### Brown adipocyte peroxisome deficiency impairs mitochondrial fission

Cre-mediated knockout of Pex16 in differentiated preadipocytes derived from BAT of *Pex16$^{Lox/Lox}$* mice resulted in the appearance of tubular mitochondria (Fig. 1a, b), confirming our previous results obtained using shRNA-mediated knockdown in brown adipocytes[20]. The formation of elongated mitochondria could potentially be due to impaired fission or increased fusion. To determine if mitochondrial fusion is affected in Pex16 deficient adipocytes, we performed fusion assays using mitochondrially-targeted photoactivatable GFP (mtPA-GFP) in brown adipocytes. In this assay, a small population of mitochondria in cells expressing mtPA-GFP are fluorescently activated and their subsequent fusion is measured by fluorescent quantification[24]. To visualize the entire mitochondrial network, the cells were stained with tetramethylrhodamine, ethyl ester (TMRE). As mitochondria fuse, the GFP signal diffuses throughout the mitochondrial network, leading to dilution of the fluorescence signal. This proceeded rapidly in control cells, indicating that mitochondrial fusion and fission events occur normally in these cells. In contrast, the knockout cells appeared to exhibit delayed fusion, presumably because their mitochondria were already elongated due to impaired fission. Quantitative analysis of the images indicated that *Pex16* knockout significantly decreases mitochondrial fusion (Fig. 1c). Thus, the appearance of tubular mitochondria in Pex16 KO adipocytes is more likely due to impaired fission rather than increased fusion.

The distribution of mtDNA within cells depends on continuous division and fusion events that are responsive to the specific needs of the cell type[25]. Replication of mtDNA is coordinated with fission events. Contacts between ER and mitochondria, which mediate mitochondrial fission, are necessary to permit mtDNA replication, though the underlying molecular mechanisms remain to be defined[26]. In addition, disruption of mitochondrial dynamics affects the integrity of mtDNA. For example, Opa1 mutants exhibit mtDNA instability[27–29]. Pex16 KO resulted in a marked decrease in mtDNA content, which was measured using a PCR-based assay to assess relative levels of the 12S mitochondrial rRNA gene (Fig.1d). Using this assay, similar results were observed in Drp1 knockdown cells[30]. Decreased mtDNA could potentially reflect increased mitophagy which sometimes follows mitochondrial fission in order to remove dysfunctional mitochondria[31]. However, assessment of mitophagy using mito-Keima, a mitochondrially-targeted pH sensitive fluorescent reporter of autophagy that exhibits green fluorescence at neutral pH and red fluorescence in the acidic environment of autophagosomes[32], suggested that Pex16 inactivation does not promote mitophagy (Supplementary Fig. 1a). Consistent with the decreased mtDNA content, Pex16 inactivation resulted in impaired mitochondrial respiration (Supplementary Fig. 1b).

Our previous studies implicated peroxisome-derived lipids called plasmalogens in mitochondrial dynamics[20]. Synthesis of these ether lipids starts in peroxisomes but is completed in the ER (Fig. 1e)[33,34]. Mimicking the effect of peroxisome deficiency, knockdown of the peroxisomal ether lipid synthetic enzyme glyceronephosphate-O-acyltransferase (GNPAT) (Fig. 1f, g) resulted in elongated mitochondria and decreased mtDNA (Fig. 1h, i). Treatment of GNPAT KD brown adipocytes with alkylglycerols (AG), plasmalogen precursors that enter the synthetic pathway downstream of the peroxisomal steps, rescued mitochondrial fragmentation and restored mtDNA content (Fig. 1h, i).

### TMEM135 is the most decreased protein in mitochondria of peroxisome-deficient BAT

To understand the mechanism through which peroxisomes regulate mitochondrial dynamics, we characterized changes in the mitochondrial proteome resulting from peroxisome deficiency (Fig. 2). To this end, we isolated mitochondria from BAT of cold-treated *Pex16-AKO* and control mice and performed mass spectrometry-based

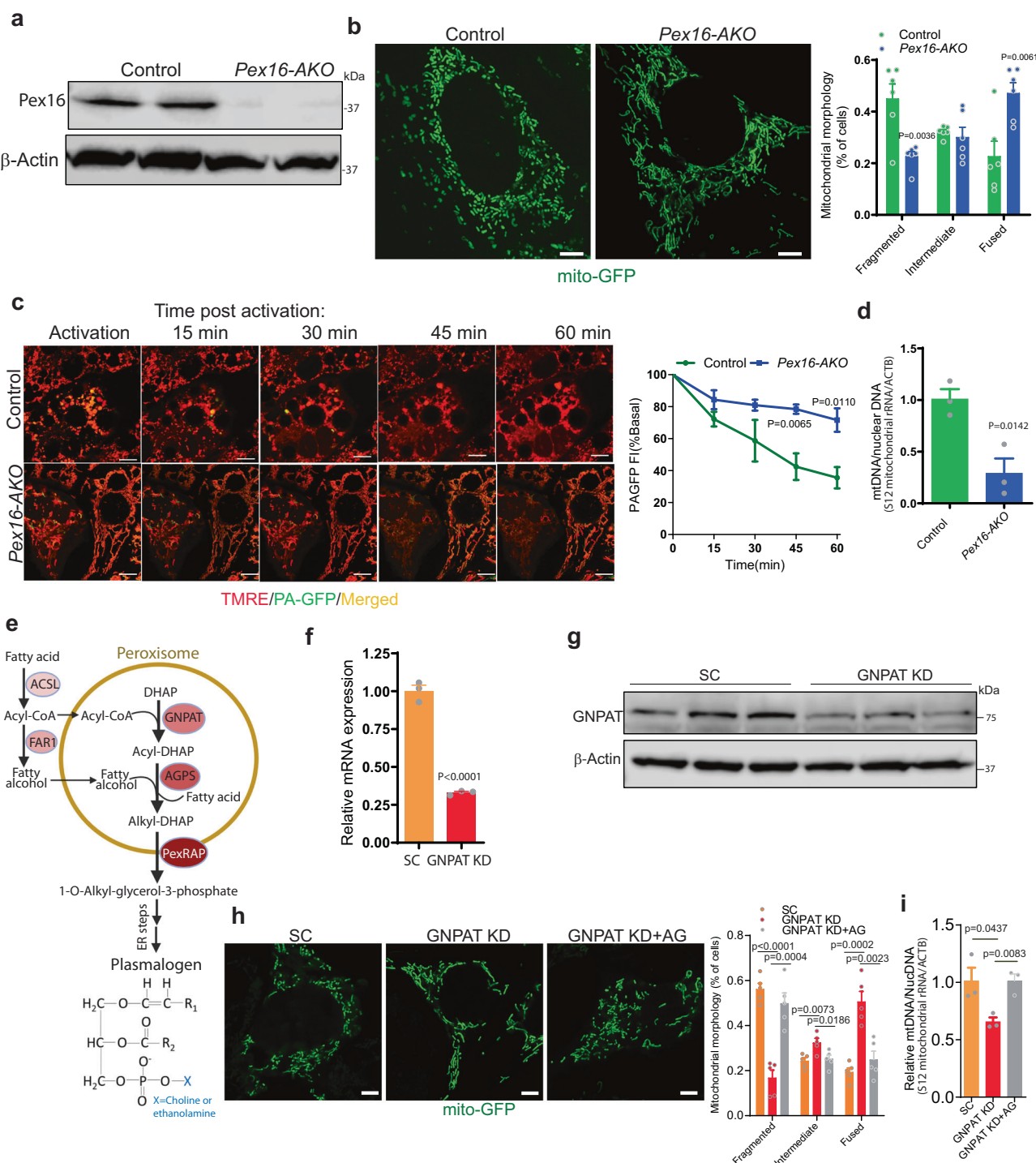

**Fig. 1 | Brown adipocyte peroxisome deficiency results in tubular mitochondrial networks through impaired mitochondrial fission, which could be rescued by peroxisome-derived lipids. a** Western blot analysis following differentiation of brown pre-adipocytes isolated from control (*Pex16lox/lox*) and *Pex16-AKO* (*Pex16lox/lox*; adipo-Cre+/-) mice. The blots are representative of three independent experiments. **b** Mitochondrial morphology of differentiated brown pre-adipocytes from control (*n* = 6) and *Pex16-AKO* (*n* = 6) mice stably expressing mito-GFP. Mitochondrial morphology was quantified using ImageJ. Scale bar: 5 μm. **c** Mitochondrial fusion assay in differentiated brown pre-adipocytes from control and *Pex16-AKO* mice transduced with lentivirus encoding mt-PAGFP, followed by confocal microscopy; control: *n* = 4; *Pex16-AKO*: *n* = 4. Mitochondrial morphology was quantified using ImageJ. Scale bar: 5 μm. **d** mtDNA copy number normalized to nuclear DNA of brown adipocytes from control (*n* = 3) and *Pex16-AKO* (*n* = 3) mice measured by qPCR. **e** Plasmalogen synthetic pathway. The initial steps for synthesis

of plasmalogens take place in peroxisomes, generating 1-*O*-alkyl-glycerol-3-phosphate, a precursor for plasmalogens synthesized in the ER. Plasmalogen structure is shown. ACSL acyl-CoA synthetase, AGPS alkylglycerone phosphate synthase, DHAP dihydroxyacetone phosphate, FAR1 fatty acyl-CoA reductase, GNPAT glycerone-phosphate O-acyltransferase, PexRAP Peroxisomal Reductase activating PPARγ. **f**, **g** qPCR and Western blot analysis of GNPAT expression in brown adipocytes treated with scrambled (SC; *n* = 3) or GNPAT shRNA (*n* = 3). **h** Mitochondrial morphology of brown adipocytes expressing Mito-GFP treated with scrambled (SC) shRNA (*n* = 5), GNPAT shRNA (*n* = 5) or GNPAT shRNA with alkylglycerol (AG; *n* = 5). Mitochondrial morphology was quantified using ImageJ. Scale bar: 5 μm. **i** mtDNA copy number normalized to nuclear DNA of brown adipocytes treated with scrambled (SC) shRNA (*n* = 3), GNPAT shRNA (*n* = 3) or GNPAT shRNA with AG (*n* = 3). Data are presented as mean ± SEM; statistical significance was determined by two-tailed unpaired Student's *t* test (**b**, **c**, **d**, **f**, **h**, **i**).

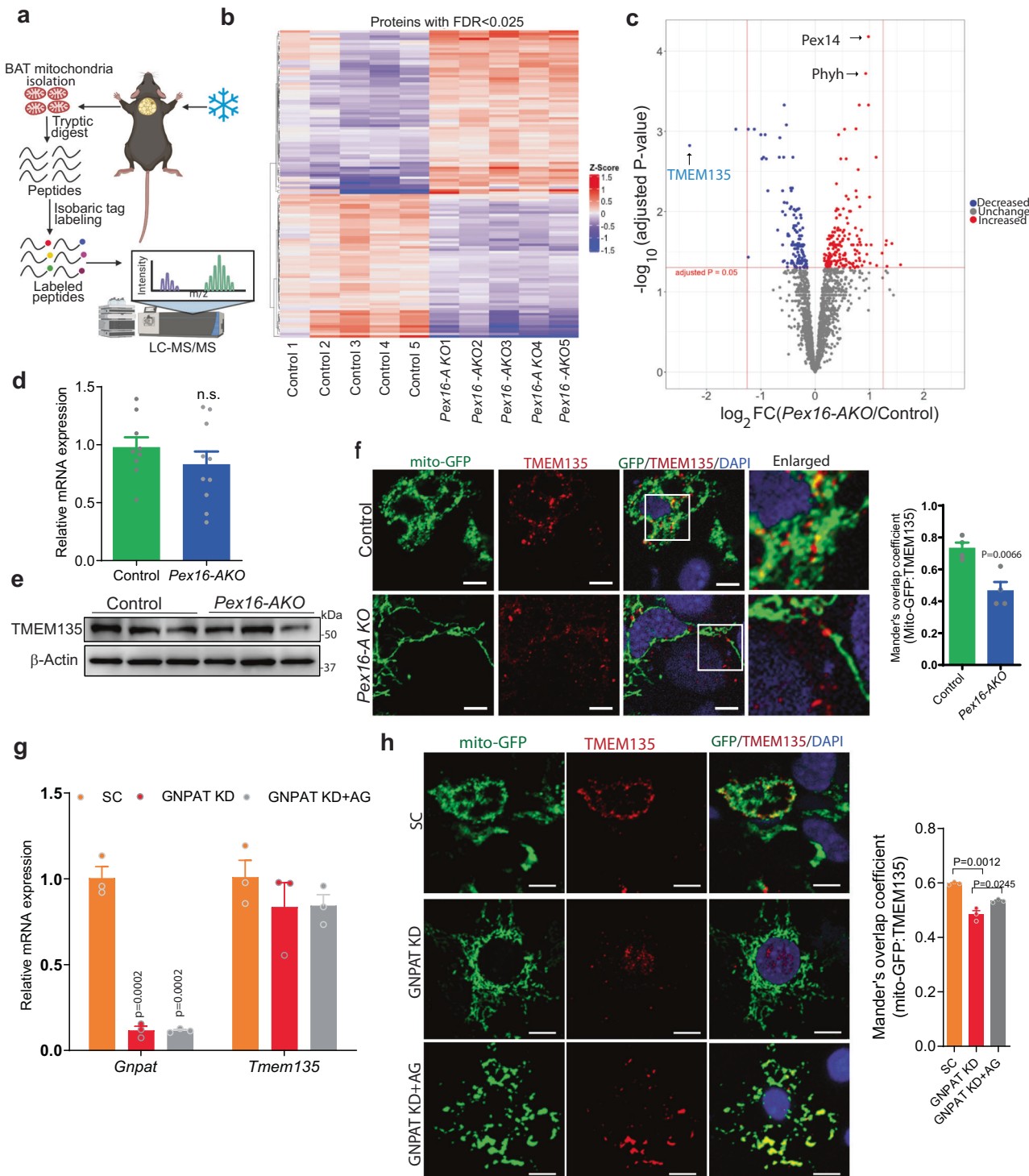

**Fig. 2 | TMEM135 is the most decreased protein in mitochondria of peroxisome-deficient BAT and its mitochondrial localization is mediated by peroxisome-derived lipids. a** Schematic diagram of proteomic analysis in mitochondria isolated from BAT of cold-treated *Pex16-AKO* and control mice. Created with BioRender.com. **b, c** Heat map and volcano plot analysis of mitochondrial proteomics in BAT of *Pex16-AKO* and control mice; *n* = 5/group. **d, e** qPCR and Western blot analysis of TMEM135 in BAT of control and *Pex16-AKO* mice after cold exposure; control: *n* = 9; *Pex16-AKO*: *n* = 10. **f** Immunofluorescence analysis in control and *Pex16-AKO* brown adipocytes expressing mito-GFP stained with an antibody against TMEM135. Colocalization was quantified using ImageJ. Quantification was performed with the investigator blinded to the identity of the samples. Scale bar: 10 μm. The images are representative of four independent experiments and the

quantification is based on a total of 54 control and 64 Pex16-AKO cells. **g** qPCR analysis of *Gnpat* and *Tmem135* in differentiated BAT SVF cells treated with lentivirus-encoding scrambled (SC) shRNA (*n* = 3), GNPAT shRNA (*n* = 3) or GNPAT shRNA plus AG (*n* = 3). **h** Immunofluorescence analysis using an antibody against TMEM135 in the brown adipocytes expressing Mito-GFP treated with lentivirus-encoding scrambled (SC), GNPAT shRNA or GNPAT shRNA plus AG. Colocalization was quantified using ImageJ. Scale bar: 10 μm. The images are representative of three independent experiments and the quantification is based on a total of 10 cells per group. Data are presented as mean ± SEM; statistical significance was determined by two-tailed unpaired Student's *t* test (**d**, **f–h**). Comparisons between groups were made with a two-tailed unpaired Student's t-test adjusted for multiple comparisons using the Benjamini-Hochberg method (**c**).

proteomics (Fig. 2a). This experiment was done in cold-treated animals since our previous studies indicate that *Pex16-AKO* mice have impaired cold-induced mitochondrial fission in BAT[20]. Since mitochondria isolated by differential centrifugation are frequently contaminated with other organelles, especially peroxisomes[24], we assessed the purity of the mitochondrial fraction by Western blot analysis using antibodies against different organelle markers. Our results show that the mitochondrial fraction was enriched in the known mitochondrial protein COX4 and had only a trace amount of the peroxisomal marker PMP70 or the nuclear marker Lamin A/C (Supplementary Fig. 2a). A complete list of the proteins identified by mass spectrometry in the two groups is presented in Supplementary Data 1. Gene Ontology (GO) analysis confirmed enrichment of mitochondrial proteins (Supplementary Fig. 2b). A heatmap of the proteins increased or decreased in the mitochondrial fraction as a result of Pex16 knockout is presented in Fig. 2b. A total of 139 proteins were significantly decreased and 182 proteins were significantly increased in mitochondria as a result of peroxisome deficiency. Increased proteins included certain peroxisomal proteins, such as the peroxisomal biogenesis factor Pex14 and the branched chain fatty acid oxidation enzyme Phyh, which were apparently mistargeted to mitochondria in the absence of peroxisomes (Fig. 2c). The mislocalization of Pex14 to mitochondria in Pex16 KO brown adipocytes was confirmed by immunofluorescence analysis (Supplementary Fig. 2c). Certain peroxisomal proteins were also decreased in *Pex16-AKO* mitochondria, including PMP70 (ABCD3) and Acox1, which our previous studies indicate are degraded in the absence of peroxisome[20]. The protein most overall decreased in the knockout mitochondria was a transmembrane protein called TMEM135 (Fig. 2c), which was originally identified by mass spectrometry as a protein enriched in peroxisomes isolated from liver and kidney and named PMP52[21,22].

The decrease in mitochondrial protein levels of TMEM135 was not associated with a change in gene expression (Fig. 2d). Moreover, the TMEM135 protein levels in whole tissue lysates were unchanged (Fig. 2e), suggesting that the decreased mitochondrial levels are not due to degradation of the protein. Confocal microscopy confirmed that mitochondrial localization of TMEM135 was reduced in brown adipocytes from *Pex16-AKO* mice (Fig. 2f). TMEM135 has been reported to be present in peroxisomes in Huh7 cells[35]. In addition, the protein is localized in mitochondria and regulates mitochondrial fission in the retina through an unclear mechanism[23]. Together, this provided a strong scientific rationale for pursuing the role of TMEM135 in the peroxisomal regulation of mitochondrial dynamics.

Analysis of the protein domain architecture indicated that TMEM135 has an N-terminal TMEM135_Cys_rich domain and a C-terminal TIM17 domain (Supplementary Fig. 2d), which suggests that the protein could be a previously uncharacterized component of the mitochondrial translocase machinery[36]. In addition, TMEM135 has five predicted transmembrane domains. Consistent with the notion that TMEM135 is localized in peroxisomes and that the import of peroxisomal membrane proteins from the cytosol where they are synthesized requires the chaperone protein Pex19[37], TMEM135 has a putative Pex19 binding motif (mPTS) (Supplementary Fig. 2d) and interacts with Pex19 (Supplementary Fig. 2e). Subcellular fractionation in brown adipocytes indicated that TMEM135 is present in peroxisomes and mitochondria, as expected, but surprisingly also in the nucleus (Supplementary Fig. 2f).

Since plasmalogen synthesis is important for the role of peroxisomes in mitochondrial fission, we wanted to determine whether these peroxisome-derived lipids regulate the mitochondrial localization of TMEM135. Knockdown of GNPAT in brown adipocytes did not affect the gene expression of *Tmem135* (Fig. 2g), but markedly influenced its subcellular localization (Fig. 2h). In control brown adipocytes, TMEM135 was localized in mitochondria, which exhibited fragmented morphology. However, knockdown of GNPAT resulted in

mislocalization of TMEM135 to the nucleus and the mitochondria had a tubular morphology. Treatment of GNPAT knockdown cells with AG rescued mitochondrial localization of TMEM135 and fragmentation of mitochondria (Fig. 2h). Together, these studies suggest that peroxisome-derived lipids facilitate mitochondrial localization of TMEM135.

## TMEM135 expression in brown adipocytes is regulated by thermogenic stimuli

TMEM135 is a poorly characterized protein. To understand its role in brown adipocytes, we wanted to characterize its gene expression profile (Fig. 3). *Tmem135* gene expression was low in undifferentiated BAT stromal vascular fractions (SVF) cells, but dramatically increased during differentiation (Supplementary Fig. 3a). Treatment of fully differentiated brown adipocytes with norepinephrine (NE), a β-adrenergic receptor agonist that promotes thermogenesis by activating lipolysis and increasing thermogenic gene expression, further increased the expression of *Tmem135* (Supplementary Fig. 3b). The transmembrane protein is expressed in multiple tissues, but its expression is high in BAT, where it further increases with prolonged cold exposure (Fig. 3a). Western blot analysis confirmed that the TMEM135 protein levels dramatically increase in BAT in mice subjected to cold exposure (Fig. 3b). Conversely, the *Tmem135* gene expression significantly decreased in BAT of mice maintained at thermoneutrality (Fig. 3c).

## TMEM135 knockdown in brown adipocytes impairs mitochondrial fission and respiration

We next determined whether TMEM135 affects mitochondrial morphology in brown adipocytes. Knockdown of TMEM135 using lentivirus-delivered shRNA in differentiated BAT SVF cells (Fig. 3d), resulted in tubular mitochondria that formed net-like structures (Fig. 3e), a feature resembling impaired mitochondrial fission caused by genetic inactivation of the fission factor Drp1[14]. Since mitochondrial DNA replication and its equal distribution to daughter cells is linked to mitochondrial division[25], the impaired fission in TMEM135 knockdown cells resulted in decreased mtDNA copy number (Fig. 3f). These results of TMEM135 inactivation on fission and mtDNA were observed in the absence of effects on the gene expression of factors involved in mitochondrial dynamics (Fig. 3g). Moreover, uncoupled OCR was significantly decreased with TMEM135 knockdown (Fig. 3h).

## Generation of adipose-specific TMEM135 knockout mice

Based on these in vitro results, we generated mice with adipose specific knockout of TMEM135 (*Tmem135-AKO*) (Fig. 4). We used CRISPR/Cas9 to create *Tmem135^{Lox/Lox}* mice on the C57BL/6 J genetic background and crossed these mice with adiponectin-Cre transgenic mice to generate *Tmem135-AKO* mice. The targeting strategy for these mice is described in Fig. 4a. Genotyping using primers flanking LoxP sites confirmed Cre-mediated recombination of the floxed alleles (Supplementary Fig. 4a). Quantitative real-time PCR demonstrated that the *Tmem135* gene was knocked out in gonadal WAT (gWAT), inguinal WAT (iWAT), and BAT, but not in the liver of *Tmem135-AKO* mice (Fig. 4b). Western blot analysis in BAT confirmed the knockout (Fig. 4c).

## Adipose-specific knockout of *TMEM135* impairs mitochondrial fission and thermogenesis and promotes diet-induced obesity and insulin resistance

Consistent with the analysis of mitochondrial morphology by confocal microscopy in cultured brown adipocytes (Fig. 3e), ultrastructural analysis in BAT from cold-treated mice showed that TMEM135 knockout results in elongated mitochondrial morphology (Fig. 4d), with significantly increased aspect ratio (Fig. 4e). Total number of mitochondria per cell was significantly lower in BAT of cold-treated

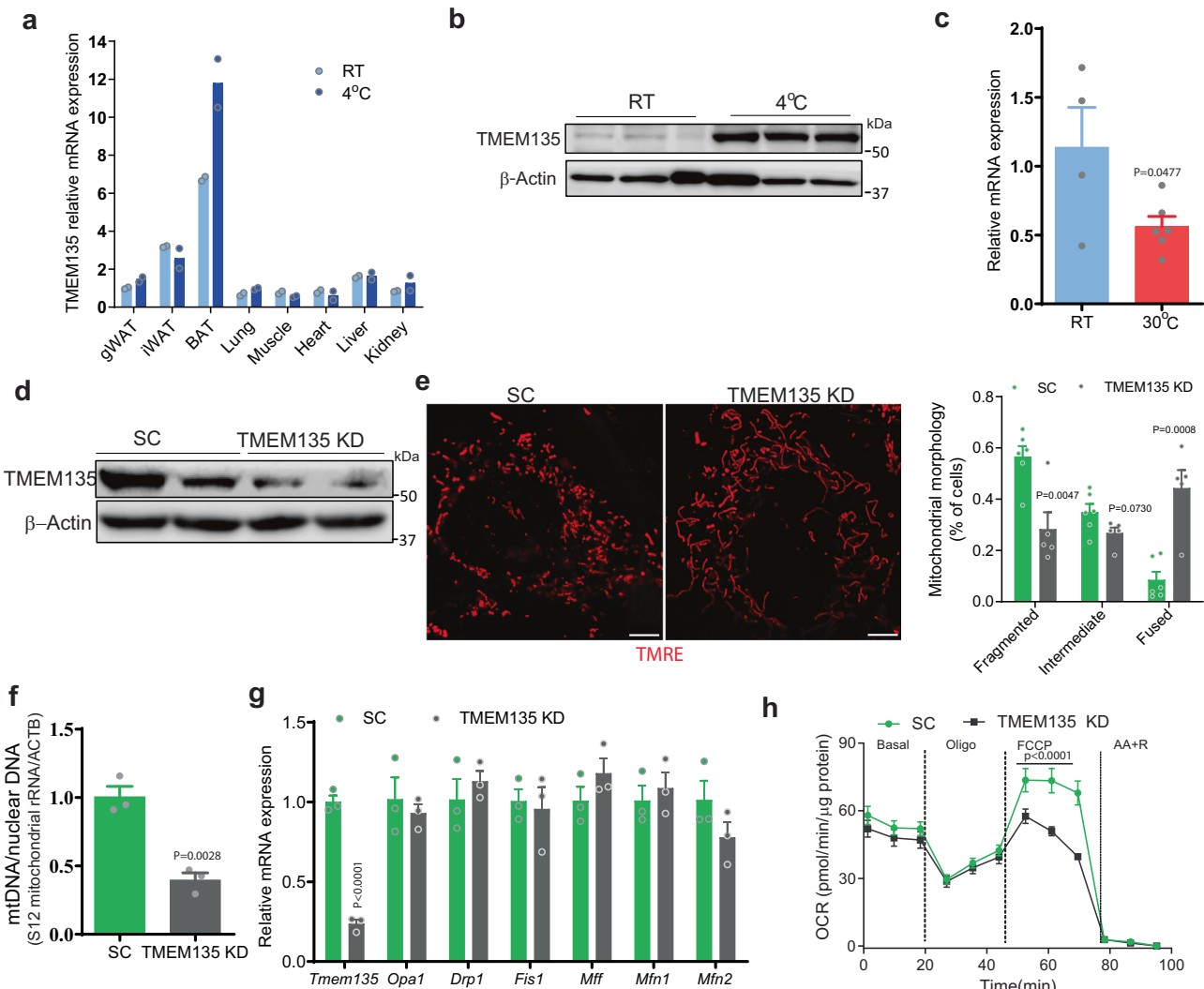

**Fig. 3 | TMEM135 expression is regulated by thermogenic stimuli and its knockdown impairs mitochondrial fission and respiration in brown adipocytes. a** qPCR analysis of *Tmem135* expression in major tissues of wild-type C57BL/6J mice maintained at room temperature or subjected to cold treatment for 7 days; $n = 2$/condition. **b** Western blot analysis of TMEM135 protein levels in BAT of wild-type C57BL/6J mice kept at room temperature or cold treated for 7 days. Each lane represents a separate mouse; $n = 3$. **c** qPCR analysis of *Tmem135* in BAT of wild-type C57BL/6J mice kept at room temperature (RT) or thermoneutrality (30 °C) for 12 days; RT: $n = 4$; 30 °C: $n = 6$. **d** Western blot analysis of TMEM135 protein levels in brown adipocytes treated with scrambled (SC) or TMEM135 shRNA. **e** Mitochondrial morphology analysis using confocal microscopy in TMRE-stained brown adipocytes treated with scrambled (SC) ($n = 6$) or TMEM135 shRNA ($n = 5$).

Plot shows quantification of mitochondrial morphology. Scale bar: 5 μm. **f** mtDNA copy number normalized to nuclear DNA of brown adipocytes treated with scrambled (SC) ($n = 3$) or TMEM135 shRNA ($n = 3$). **g** qPCR analysis of mitochondrial dynamics genes in brown adipocytes treated with scrambled (SC) ($n = 3$) or TMEM135 shRNA ($n = 3$). **h** OCR measured in brown adipocytes treated with scrambled (SC) ($n = 5$) or TMEM135 shRNA ($n = 3$) using a Seahorse XF24 Extracellular Flux Analyzer. Oligo oligomycin, FCCP carbonyl cyanide-p-trifluoromethoxyphenylhydrazone, AA + R, mixture of antimycin A and rotenone. Data are presented as mean ± SEM; statistical significance was determined by two-tailed unpaired Student's *t* test (**c**, **e–g**) or 2-way ANOVA with Bonferroni's post hoc test (**h**).

*Tmem135-AKO* mice as compared to control animals (Fig. 4f). Gene expression analysis indicated that TMEM135 KO brown adipocytes have significantly decreased gene expression of the mtDNA-encoded genes *MtCO1* and *MtND6*, while nuclear-encoded genes for mitochondrial proteins (e.g. *Ucp1*) and factors involved in mitochondrial biogenesis (*Tfam* and Nrf1) were unchanged (Supplementary Fig. 4b). Assessment of mitochondrial bioenergetics using a Seahorse Extracellular Flux Analyzer in isolated BAT mitochondria revealed that TMEM135 KO decreases OCR (Supplementary Fig. 4c). To determine if the altered BAT mitochondrial morphology affects thermogenesis, we next assessed the effect of TMEM135 knockout on cold tolerance. Both male and female *Tmem135-AKO* mice were significantly less cold tolerant than control mice (Fig. 4g and Supplementary Fig. 4d). Moreover, β3-adrenoreceptor-dependent O₂ consumption (VO₂) was

significantly decreased in *Tmem135-AKO* mice (Fig. 4h), suggesting that adipose tissue TMEM135 is important for thermogenesis in vivo. Accordingly, BAT of cold-treated *Tmem135-AKO* mice exhibited a marked intracellular accumulation of lipids (Fig. 4i).

To determine the effect of TMEM135 inactivation on adiposity and metabolism, we phenotypically characterized the *Tmem135-AKO* mice. High-fat feeding (60 kcal% fat) resulted in significantly higher body weight in female *Tmem135-AKO* mice compared to control animals (Fig. 4j), despite no difference in food intake (Supplementary Fig. 4e). Assessment of glucose homeostasis in these animals revealed that the TMEM135 knockout results in glucose intolerance, hyperinsulinemia and insulin resistance (Fig. 4k–m). In contrast to female mice, high fat feeding in male mice did not elicit a significant body weight difference between the genotypes, presumably because the already rapid

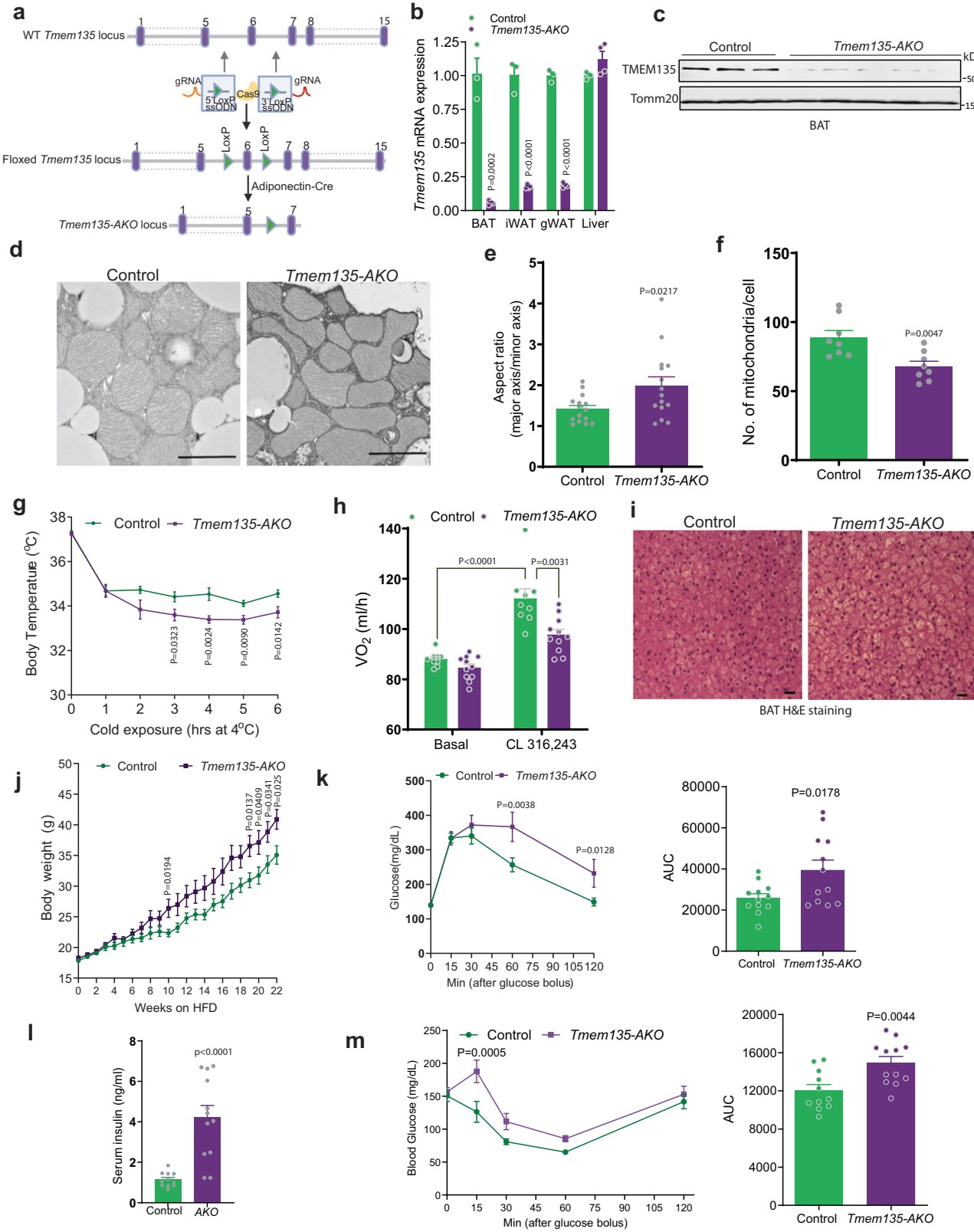

diet-induced weight gain in male C57BL/6J mice could not be further increased by the TMEM135 inactivation (Supplementary Fig. 2f). However, despite the lack of difference in body weight, HFD-fed *Tmem135-AKO* male mice exhibited glucose intolerance and insulin resistance (Supplementary Fig. 4g, h), suggesting that adipose-specific TMEM135 inactivation has a primary effect on insulin sensitivity, independent of body weight difference.

## TMEM135 overexpression increases mitochondrial fission and thermogenesis, decreases diet-induced obesity, and promotes insulin sensitivity

We next determined the effect of TMEM135 overexpression on mitochondrial morphology in brown adipocytes (Fig. 5). Whereas TMEM135 knockdown in cultured brown adipocytes resulted in tubular mitochondria (Fig. 3f), the overexpression (Fig. 5a)

**Fig. 4 | Adipose-specific knockout of TMEM135 impairs mitochondrial fission and thermogenesis and promotes diet-induced obesity and insulin resistance. a** Gene targeting strategy using CRISPR/Cas9 to insert LoxP sites into the *Tmem135* locus. The floxed mice were crossed with an adiponectin-Cre mouse to generate *Tmem135-AKO* mice. **b** qPCR analysis of *Tmem135* expression in adipose tissue and liver of control (*n* = 3) and *Tmem135-AKO* (*n* = 4) mice. **c** Western blot analysis of TMEM135 in brown adipose tissue of control (*n* = 3) and *Tmem135-AKO* (*n* = 5) mice. **d** TEM analysis of BAT from control and *Tmem135-AKO* mice subjected to 7 day cold exposure. Scale bar: 1 µm. **e** Aspect ratio (ratio of major axis length to minor axis length) measured in BAT mitochondria. The data are based on 15 mitochondria per condition. **f** Number of mitochondria per cell based on TEM images of BAT taken at ×1000–×2000 magnification. Data are the average of 8 cells per condition. **g** Body temperature of *Tmem135-AKO* (*n* = 10) and control (*n* = 9) female mice subjected to a 6-h cold challenge. (**h**) VO₂ of *Tmem135-AKO* (*n* = 11) and control (*n* = 9) mice treated with β3-adrenergic agonist CL-316,243 injection. **i** H&E staining of BAT from *Tmem135-AKO* and control mice subjected to cold exposure. The images are representative of 3 mice per genotype. Scale bar: 50 µm. **j** Body weight of HFD-fed *Tmem135-AKO* and control female mice; *n* = 12/group. **k** Glucose tolerance test and area under the curve (AUC) of HFD-fed TMEM135-AKO and control female mice; *n* = 12/group. **l** Fasting serum insulin level of HFD-fed *Tmem135-AKO* (*n* = 12) and control (*n* = 11) female mice. **m** Insulin tolerance test and AUC of HFD-fed *Tmem135-AKO* (*n* = 12) and control (*n* = 11) female mice. Data are presented as mean ± SEM; statistical significance was determined by two-tailed unpaired Student's *t* test (**b**, **e**–**h**, **j**–**m**) or 2-way ANOVA with Bonferroni's post hoc test (**k**, **m**).

increased mitochondrial fragmentation (Fig. 5b) and mtDNA content (Fig. 5c).

Transgenic mice that globally overexpress TMEM135 (*Tmem135^TG*) under the control of CAG promoter exhibit increased mitochondrial fragmentation in the retina and the heart[23,38]; however the underlying molecular mechanisms remain unclear. Moreover, the effects of TMEM135 overexpression on thermogenesis and whole-body energy metabolism have not been investigated. TMEM135 was overexpressed five- to ten-fold in the different adipose tissue depots of the transgenic mice (Fig. 5d). Since mitochondrial fission in brown adipocytes is associated with increased uncoupled respiration[8,12,20], we next determined the effect of TMEM135 overexpression on thermogenesis. *Tmem135^TG* mice adapted to thermoneutrality for 12 days and then subjected to an acute cold challenge were markedly more cold tolerant than similarly treated wild-type (WT) mice (Fig. 5e), with no change in gene expression of proteins involved in mitochondrial dynamics or thermogenesis, including UCP1 (Supplementary Fig. 5a).

We also metabolically phenotyped male and female *Tmem135^TG* mice and similar results were obtained with both sexes. Although *Tmem135^TG* mice fed normal chow diet lacked a body weight difference, the body weight was significantly decreased in the HFD-fed transgenic mice (Fig. 5f), despite no difference in food intake (Supplementary Fig. 5b) or locomotor activity (Supplementary Fig. 5c) compared to WT mice. In addition, fecal fat content was not different between the genotypes (Supplementary Fig. 5d), indicating that the decreased diet-induced obesity in the transgenic mice is not due to malabsorption. Resting energy expenditure (VO₂) measured at 22 °C in HFD-fed mice prior to emergence of a body weight difference trended higher in *Tmem135^TG* mice, but the effects did not reach statistical significance (Supplementary Fig. 5e, f), reflecting the fact that indirect calorimetry is not sufficiently sensitive to detect subtle changes in energy expenditure that impact body weight over a prolonged period[39,40]. However, carbon dioxide production (VCO₂) and respiratory exchange ratio (RER) were significantly higher in *Tmem135^TG* mice compared to WT animals (Supplementary Fig. 5f, g). To examine changes in energy utilization at thermoneutrality, we performed real-time measurements of ambient CO₂ in mice maintained at 30 °C for 5 days. CO₂ was measured using a wireless sensor capable of measuring multiple parameters including activity, light, temperature, humidity, and CO₂ levels from within the mouse home-cages (Supplementary Fig. 5h). CO₂ from mouse respiration accumulates in the cage, resulting in a CO₂ level of ~2000–3000ppm for singly housed mice, ~5 times the concentration of CO₂ in room air. Consistent with a decrease in metabolic rate at thermoneutrality[41], ambient CO₂ in cages with WT mice progressively decreased over the 5 days of this experiment (Supplementary Fig. 5h). In contrast, CO₂ measured from cages containing mice that overexpressed TMEM135 did not reduce, and after 5 days the ambient CO₂ in TME135 overexpressing cages was significantly higher than cages with WT mice (Supplementary Fig. 5h).

Body composition analysis using EcoMRI in HFD-fed mice showed a significantly decreased fat mass and no difference in lean mass in the *Tmem135^TG* mice (Fig. 5g). Histologic examination showed that TMEM135 overexpression results in markedly smaller adipocytes in the iWAT depot and a striking protection against obesity-associated whitening of BAT (Fig. 5h). Moreover, high fat feeding resulted in a massive fatty liver in WT mice, but the liver appeared normal in *Tmem135^TG* mice (Fig. 5i). Accordingly, HFD-fed *Tmem135^TG* mice exhibited a marked improvement in glucose tolerance (Fig. 5j and Supplementary Fig. 5i) in the context of hypoinsulinemia (Fig. 5k) and increased circulating adiponectin levels (Supplementary Fig. 5j), suggesting that TMEM135 promotes insulin sensitivity. The increased insulin sensitivity was confirmed using insulin tolerance testing (ITT) (Fig. 5l and Supplementary Fig. 5k) and Western blot analysis of insulin-stimulated Akt phosphorylation in liver (Fig. 5m). Since the transgenic mice globally overexpress TMEM135, we wanted to determine if increased glucose uptake by BAT and/or other metabolic tissues is involved in mediating the improved glucose homeostasis caused by TMEM135 overexpression. To this end, we performed ¹⁸F-FDG PET/CT scan on HFD-fed mice. Quantification of ¹⁸F-FDG uptake indicated that TMEM135 overexpression selectively increased glucose uptake in BAT, while there was no difference in the glucose uptake normalized to tissue weight between WT and *Tmem135^TG* mice in other tissues, such as iWAT, gWAT, skeletal muscle and liver (Supplementary Fig. 5l). The increased glucose uptake in transgenic BAT is consistent with a more thermogenically active brown fat with markedly decreased intracellular lipid content (Fig. 5h).

## TMEM135 overexpression in *Pex16-AKO* mice rescues BAT mitochondrial fission and cold tolerance

We next wanted to explore the hypothesis that the peroxisome deficiency-associated defect in mitochondrial fission and cold tolerance is due to decreased mitochondrial protein levels of TMEM135. To test this hypothesis, we first determined the effect of overexpressing TMEM135 in Pex16 knockout brown adipocytes on mitochondrial morphology (Supplementary Fig. 6a). Whereas Pex16 knockout cells exhibited elongated mitochondrial morphology, overexpression of TMEM135 in the context of *Pex16* genetic inactivation resulted in fragmented mitochondria. Immunofluorescence analysis using an anti-TMEM135 antibody confirmed that the overexpressed TMEM135 was able to at least partially localize to mitochondria in these brown adipocytes, despite peroxisome deficiency (Supplementary Fig. 6a). To translate the effect on mitochondrial morphology to mice, we crossed *Tmem135^TG* mice with *Pex16-AKO* mice to generate *Pex16-AKO/Tmem135^TG* mice (Supplementary Fig. 6b). Quantitative real-time PCR and Western blot analysis confirmed that Pex16 was knocked out and TMEM135 was overexpressed in *Pex16-AKO/Tmem135^TG* mice (Supplementary Fig. 6c, d). TEM analysis indicated that mitochondria in the BAT of Pex16-AKO exhibit an elongated morphology after cold treatment, consistent with our previously reported results[20]. Notably, TMEM135 overexpression in *Pex16-AKO* mice rescued the mitochondrial

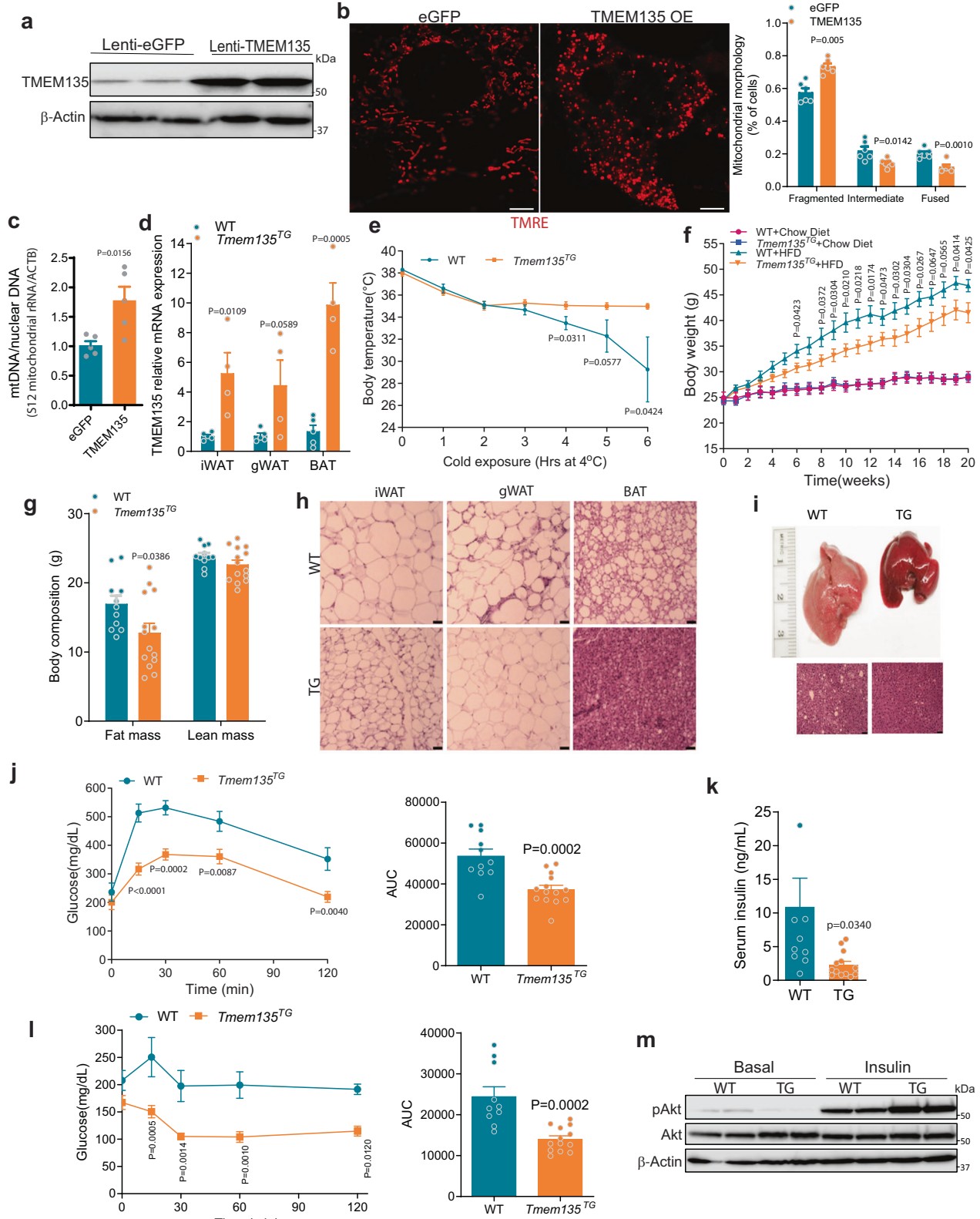

morphology, resulting in circular mitochondrial (Supplementary Fig. 6e) with an aspect ratio of close to 1 (Supplementary Fig. 6f) and restored the mtDNA content in BAT (Supplementary Fig. 6g). These data indicate that overexpression of TMEM135 rescues the impaired mitochondrial fission resulting from peroxisome deficiency.

We next determined the effect of TMEM135 overexpression on cold tolerance. Although *Pex16-AKO* mice were severely cold

intolerant, overexpression of TMEM135 in the Pex16-deficient mice significantly improved cold tolerance (Supplementary Fig. 6h) and decreased intracellular accumulation of lipids in BAT (Supplementary Fig. 6i). Taken together, these findings suggest that TMEM135 is a critical downstream mediator of the peroxisomal regulation of mitochondrial fission and thermogenesis.

**Fig. 5 | TMEM135 overexpression increases mitochondrial fission and thermogenesis, decreases diet-induced obesity, and promotes insulin sensitivity.**
**a** Western blot analysis of TMEM135 protein levels in brown adipocytes transduced with lentivirus expressing GFP or TMEM135. The blots are representative of three independent experiments. **b** Mitochondrial morphology analysis using confocal microscopy in TMRE-stained of brown adipocytes expressing GFP ($n = 6$) or TMEM135 ($n = 6$). Plot shows quantification of mitochondrial morphology. Scale bar: 5 μm. **c** mtDNA copy number normalized to nuclear DNA of brown adipocytes expressing GFP ($n = 5$) or TMEM135 ($n = 5$). **d** qPCR analysis of *Tmem135* in various adipose depots of *Tmem135*$^{TG}$ ($n = 4$) and WT ($n = 5$) mice. **e** Body temperature of *Tmem135*$^{TG}$ ($n = 7$) and WT ($n = 5$) mice kept in warm room (30 °C) for 12 days and then subjected to a 6-h cold challenge. **f** Body weight of chow diet or HFD-fed *Tmem135*$^{TG}$ and WT male mice; chow diet (WT: $n = 5$; *Tmem135*$^{TG}$: $n = 5$) and HFD (WT: $n = 11$; *Tmem135*$^{TG}$: $n = 14$). **g** EchoMRI analysis of body composition in HFD-fed

*Tmem135*$^{TG}$ ($n = 14$) and WT ($n = 11$) male mice. **h** H&E staining of adipose tissue depots in HFD-fed *Tmem135*$^{TG}$ and WT mice. Scale bar: 50 μm. The images are representative results of three independent experiments. **i** Gross images and H&E staining of liver in HFD-fed *Tmem135*$^{TG}$ and WT male mice. Scale bar: 50 μm. The images are representative of three mice per group. **j** Glucose tolerance testing and AUC in HFD-fed *Tmem135*$^{TG}$ ($n = 14$) and WT ($n = 11$) male mice. **k** Fasting serum insulin level of HFD-fed *Tmem135*$^{TG}$ ($n = 13$) and WT ($n = 10$) male mice. **l** Insulin tolerance testing and AUC in HFD-fed *Tmem135*$^{TG}$ ($n = 13$) and WT ($n = 10$) male mice. **m** Western blot analysis of AKT phosphorylation in livers of HFD-fed *Tmem135*$^{TG}$ and WT mice collected at 10 min following insulin injection. Each lane represents a separate mouse. Two independent experiments yielded similar results. Data are presented as mean ± SEM; statistical significance was determined by two-tailed unpaired Student's *t* test (**b**–**g**, **k**) or 2-way ANOVA with Bonferroni's post hoc test (**j**, **l**).

## Acute β-adrenergic activation promotes association between peroxisomes and mitochondria and translocation of TMEM135 to mitochondria

Mitochondrial fission involves recruitment of the ER to the constriction site marked by mtDNA. Subsequently, outer membrane-bound proteins such as Fis1 and MFF recruit Drp1 to the surface of the mitochondria to assist in ER-mediated constriction[42]. Given our results indicating that peroxisomes are important for mitochondrial fission, we determined whether peroxisomes are recruited to mitochondria in response to β-adrenergic activation (Fig. 6), which promotes mitochondrial fission in brown adipocytes. Live cell imaging in brown adipocytes expressing peroxisome-targeted GFP (GFP-PTS1) and stained with TMRE to label mitochondrial networks indicated that NE treatment promotes colocalization of peroxisomes with mitochondria (Fig. 6a), suggesting that peroxisomes are recruited to mitochondria under conditions that promote mitochondrial division.

Since TMEM135 is a critical mediator of the peroxisomal regulation of mitochondrial fission, we next determined whether it translocates from peroxisomes to mitochondria during mitochondrial fission. To that end, we labelled mitochondria with TMRE in brown adipocytes expressing GFP-PTS1 and blue fluorescent protein (BFP)-tagged TMEM135, and monitored TMEM135 subcellular localization using two-photon confocal microscopy after treatment with NE. In control treated cells, TMEM135 exhibited colocalization with the GFP-PTS1. After NE treatment, the colocalization with the mitochondrial maker significantly increased (Fig. 6b), suggesting that TMEM135 translocates from peroxisomes to mitochondria during conditions that promote mitochondrial fission. Consistent with this possibility, subcellular fractionation analysis in brown adipocytes indicated that the mitochondrial presence of TMEM135 increases in response to NE treatment (Fig. 6c).

## TMEM135 interacts with AKAP1 and promotes PKA-dependent phosphorylation of Drp1

We next investigated the molecular mechanism through which TMEM135 promotes mitochondrial fission. Mitochondrial fragmentation requires Drp1, which is recruited to the mitochondrial outer membrane during fission[13]. Mitochondrial translocation of Drp1 in brown adipocytes requires PKA-mediated Ser600 phosphorylation of the GTPase in response to β-adrenergic stimulation[8]. CRISPR/Cas9-mediated knockout of TMEM135 inhibited Ser600 phosphorylation of mouse Drp1 (Ser637 in humans) in NE-treated brown adipocytes (Fig. 6d). Conversely, TMEM135 overexpression promoted this phosphorylation event (Fig. 6e). Consistent with the notion that this phosphorylation promotes mitochondrial retention of Drp1 in certain cell types, including podocytes and brown adipocytes[8,43], confocal microscopy showed that TMEM135 overexpression increased Drp1 localization in mitochondria (Fig. 6f). These data suggest that TMEM135 is necessary and sufficient to promote mitochondrial fission through its ability to recruit the fission factor Drp1.

To gain a further understanding of the molecular mechanism through which TMEM135 promotes Drp1 phosphorylation, we searched for its interacting proteins. Recently, a high throughput proximity-dependent biotinylation approach to profile protein-protein interactions in HEK293 cells has been reported[44]. Search of this dataset revealed several proximity interactors of TMEM135 (Fig. 6g), including peroxisomal proteins acyl-CoA binding domain 5 (ACBD5), Pex3 and PXMP2, supporting our notion that TMEM135 is a peroxisome-localized protein. Importantly, A-kinase anchor protein 1 (AKAP1; also named AKAP121), a mitochondrial protein that recruits PKA to the outer mitochondrial membrane[45], was also identified as a potential TMEM135-interacting protein (Fig. 6g). Since AKAP1 has been reported to regulate mitochondrial fission by mediating PKA-dependent phosphorylation of Ser637 in Drp1[46], we determined whether it physically interacts with TMEM135. Using co-immunoprecipitation and GST-pulldown assays, we validated the interaction of AKAP1 with TMEM135 (Fig. 6h, i). Together, these data suggest that TMEM135 is part of a multivalent complex that mediates mitochondrial fission by promoting PKA-dependent phosphorylation of Drp1.

## TMEM135 regulates mitochondrial fission in human thermogenic adipocytes

We next sought to evaluate the translational relevance of these observations to humans (Fig. 7). In humans, TMEM135 is also most highly expressed in adipose tissue among multiple other organs (Supplementary Fig. 7a), according to analysis of a publicly available dataset[47]. Similar to NE-stimulated increase in the expression in mouse brown adipocytes (Supplementary Fig. 3b), forskolin treatment increased the expression of TMEM135 in adipocytes derived from human BAT (hBAT) (Fig. 7a), suggesting that the control of TMEM135 levels in mouse and human adipocytes is downstream of cAMP-mediated thermogenic signaling. CRISPR/Cas9-mediated knockout of TMEM135 in hBAT cells (Fig. 7b) resulted in the appearance of elongated mitochondria (Fig. 7c), without affecting gene expression of proteins involved in mitochondrial fission or fusion (Supplementary Fig. 7b). Consistent with disruption of mitochondrial fission, TMEM135 KO in human brown adipocytes decreased mtDNA content (Fig. 7d), with concomitant impairment of mitochondrial respiration (Fig. 7e). Moreover, shRNA-mediated knockdown of TMEM135 in MIN6 mouse insulinoma cells (Supplementary Fig. 7c) or H9C2 rat cardiomyocytes (Supplementary Fig. 7d) promoted the formation of elongated mitochondria. Together, these data suggest that TMEM135 broadly regulates mitochondrial fission in different cell types.

Suggesting that TMEM135 might be relevant to obesity in humans, analysis of publicly available transcriptome-profiling data (GEO: GSE94753) indicates that *Tmem135* gene expression is significantly decreased in abdominal subcutaneous WAT of female subjects with obesity as compared to lean individuals (Fig. 7f). Moreover, analysis of AMP-T2D-GENES exome sequencing data of samples from Starr County, Texas[48] led to the identification of a coding variant of

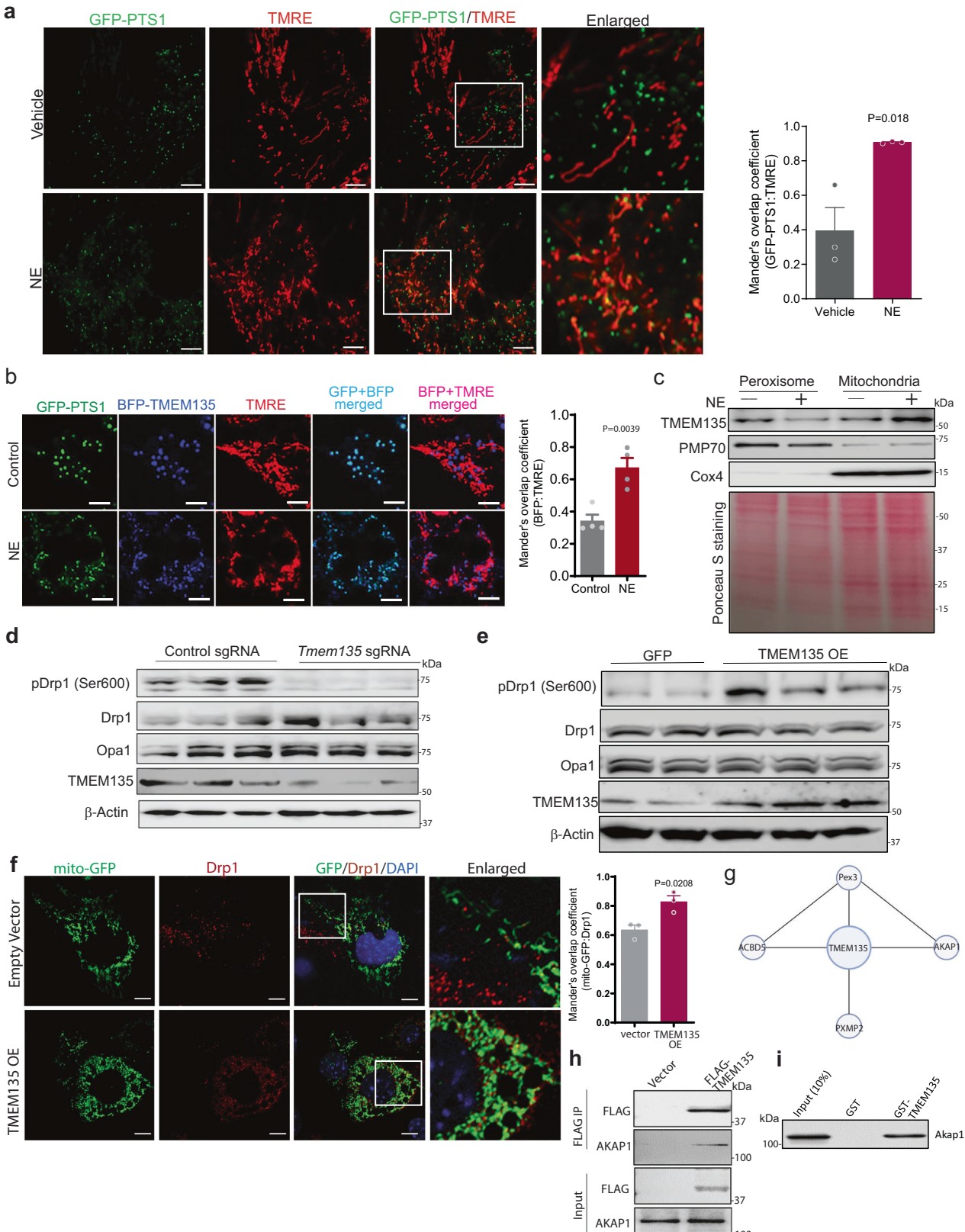

TMEM135 (dbSNP: rs549195058) that results in a glycine to glutamic acid replacement at position 369 (G369E) in transcript variant 1 (G347E in transcript variant 2) and is associated with increased BMI specifically in a sub-cohort representing 13,000 Hispanic samples (Fig. 7g). This mutation is located in a highly conserved C-terminal region of TMEM135 (Fig. 7h) adjacent to the TIM17 domain. To assess the

functional impact of this variant on mitochondrial fission, we ectopically expressed mouse WT or G369E mutant in TMEM135 knockout hBAT cells and determined the effect on mitochondrial morphology. Western blot analysis confirmed that the WT and G369E mutant TMEM135 were equally expressed in the knockout cells (Fig. 7i). Whereas TMEM135 knockout resulted in elongated mitochondria,

**Fig. 6 | TMEM135 translocates from peroxisomes to mitochondria in response to acute β-adrenergic stimulation and interacts with AKAP1 to mediate PKA-dependent phosphorylation of Drp1. a** Confocal microscopy in control ($n = 3$) or 1 h NE-treated brown adipocytes ($n = 3$) expressing peroxisome-targeted GFP (GFP-PTS1) and labelled with TMRE to mark mitochondrial networks. Quantification of peroxisome-mitochondria colocalization using ImageJ. Quantification was performed with the investigator blinded to the identity of the samples. Scale bar: 5 μm. **b** Two-photon confocal microscopy in control ($n = 4$) or 1 h NE-treated brown adipocytes ($n = 4$) expressing GFP-PTS1 and BFP-TMEM135 and stained with TMRE. Quantification of BFP-TMEM135 localization with mitochondria using ImageJ is shown. Scale bar: 5 μm. **c** Western blot analysis of TMEM135 protein levels in peroxisomal and mitochondrial fractions of NE-treated brown adipocytes. Ponceau S staining as loading control. The blots are representative of three independent experiments. **d** Western blot analysis of Drp1 (Ser600) phosphorylation in NE-treated brown adipocytes with or without CRISPR/Cas9-mediated knockout of

TMEM135; $n = 3$. **e** Western blot analysis of Drp1 (Ser600) phosphorylation in brown adipocytes transduced with lenti-GFP ($n = 2$) or lenti-TMEM135 ($n = 3$). **f** Immunofluorescence analysis using an antibody against Drp1 in mito-GFP expressing brown adipocytes transduced with empty or TMEM135 overexpression lentivirus. Colocalization was quantified using ImageJ. Scale bar: 10 μm. The images are representative of three independent experiments and the quantification is based on a total of 16 empty vector and 23 TMEM135 OE cells cells per group. **g** TMEM135 interaction network based on search of a publicly available dataset of global protein–protein interactions using a proximity-dependent biotinylation approach. **h** Co-immunoprecipitation of endogenous AKAP1 with FLAG-TMEM135 in HEK293 cells. The blots are representative of three independent experiments. **i** Pulldown assay showing that GST-TMEM135 interacts with endogenous AKAP1 in brown adipocyte protein lysates. The blots are representative of three independent experiments. Data are presented as mean ± SEM; statistical significance was determined by two-tailed unpaired Student's $t$ test (**a**, **b**, **f**).

re-expression of WT, but not G369E TMEM135, promoted the appearance of fragmented mitochondria (Fig. 7j). These data suggest that this TMEM135 variant is a loss-of-function mutant that impairs mitochondrial fission.

## Discussion

These studies identify TMEM135 as a critical mediator of the peroxisomal regulation of mitochondrial dynamics and energy homeostasis. TMEM135 is the most decreased protein in the mitochondria of adipose-specific peroxisome deficient (*Pex16-AKO*) mice. TMEM135 expression specifically increases in BAT in response to prolonged cold exposure or NE treatment. Knockdown of TMEM135 results in elongated mitochondria and decreases mitochondrial DNA copy number and respiration. Adipose-specific knockout of TMEM135 in mice blocks cold-induced BAT mitochondrial fission, impairs thermogenesis, and increases high fat diet-induced obesity and insulin resistance. Conversely, TMEM135 overexpression promotes mitochondrial fragmentation, protects against diet-induced obesity and insulin resistance, and rescues thermogenesis in *Pex16-AKO* mice. Mechanistically, thermogenic stimuli promote association between peroxisomes and mitochondria and plasmalogen-dependent translocation of TMEM135 to mitochondria, where it interacts with AKAP1 to regulate PKA-dependent phosphorylation of Drp1 and mediate mitochondrial fission, resulting in increased uncoupled respiration and thermogenesis (Fig. 8).

Mitochondrial fission involves interaction of mitochondria with other organelles[17]. Fission begins with recruitment of the ER to the constriction site, marked by mtDNA. Subsequently, Drp1 is recruited to the surface of mitochondria to assist in ER-mediated constriction[42]. Recent studies show that lysosomes and trans Golgi network (TGN) vesicles are also present at mitochondrial fission sites induced by the ER, resulting in multi-organelle contact sites during the fission process[49,50]. With the recruitment of peroxisomes to mitochondria under conditions that promote mitochondrial fission, our results uncover another layer of complexity in the organelle interactions orchestrating mitochondrial division. Importantly, our high-resolution microscopy studies shows that mitochondrial localization of TMEM135 increases in response to acute NE treatment, revealing a previously unidentified recruitment of a peroxisomal protein to mitochondria to mediate mitochondrial fission. Since de novo formation of peroxisomes involves fusion of pre-peroxisomal vesicles derived from the ER and mitochondria[51], it is conceivable that the increased mitochondrial localization of TMEM135 during mitochondrial fission reflects fusion of peroxisomal membrane with mitochondria.

Such fusion events can potentially be mediated by plasmalogens, peroxisome-derived lipids that promote mitochondrial localization of TMEM135. Ethanolamine (PE) plasmalogens are thought to facilitate rapid membrane fusion due to the presence of a vinyl ether bond at the sn-1 position of their glycerol backbone, which promotes the

formation of non-lamellar lipid structures[52,53]. PE plasmalogens are decreased in BAT mitochondria of *Pex16-AKO* mice and dietary supplementation of these lipids rescues the impaired mitochondrial fission and thermogenesis in the peroxisome-deficient mice[20], suggesting that peroxisomes channel plasmalogens to mitochondria in brown adipocytes to regulate mitochondrial fission by facilitating mitochondrial localization of TMEM135. Surprisingly, disruption plasmalogen synthesis results in nuclear translocation of TMEM135. Physiological significance of the nuclear localization is unknown but might reflect a potential role for the protein in mitochondria-to-nucleus signaling. Additional work is needed to investigate this possibility.

Drp1 is necessary, but not sufficient, for mitochondrial fission, as it must be recruited to the outer mitochondrial membrane by its receptors, such as Fis1, Mff and miD49/51[54,55]. In humans, the mitochondrial fission activity of Drp1 is regulated by its phosphorylation at Ser616 and Ser637 (corresponding to Ser579 and Ser600 in mice, respectively). PINK1-mediated phosphorylation at Ser616 promotes mitochondrial translocation and fission activity of Drp1[56]. However, the effects of Ser637 phosphorylation on mitochondrial fission appear to be cell context dependent[57] and might be influenced by the Ser616 phosphorylation status[58]. In brown adipocytes, PKA-mediated Ser637 phosphorylation in response to β-adrenergic stimulation promotes mitochondrial retention and fission activity of Drp1[8]. This phosphorylation event is also associated with increased mitochondrial fission in podocytes[43]. Our results show that genetic inactivation of *Tmem135* in brown adipocytes inhibits Drp1 Ser637 phosphorylation and results in elongated mitochondria, while the overexpression promotes Drp1 phosphorylation, leading to fragmented mitochondria. Thus, TMEM135 is a unique protein that is necessary and sufficient for mitochondrial fission.

PKA-mediated phosphorylation of Drp1 Ser637 requires AKAP1, an outer mitochondrial membrane-localized scaffolding protein that recruits PKA to the surface of mitochondria[45]. AKAP1 mediates PKA-dependent phosphorylation of Ser637 in Drp1 and promotes mitochondrial fission in podocytes[46]. Our results demonstrate that TMEM135 physically interacts with AKAP1, suggesting that it is a component of a complex involved in promoting PKA-dependent phosphorylation and mitochondrial retention of Drp1, leading to mitochondrial fragmentation.

Mitochondria alter their morphology depending on metabolic context[9,59]. Elongated mitochondria are associated with conditions requiring increased mitochondrial ATP synthesis, such as starvation[60,61]. In contrast, fragmented mitochondria are associated with reduced bioenergetics efficiency, such as nutrient surplus or uncoupled respiration[62]. In brown adipocytes, circular mitochondria resulting from division of the organelle exhibit increased uncoupled respiration[8]. Our results demonstrate that TMEM135 overexpression promotes mitochondrial fission, uncoupled respiration and thermogenesis, while the genetic inactivation of *Tmem135* leads to the opposite effects. Together,

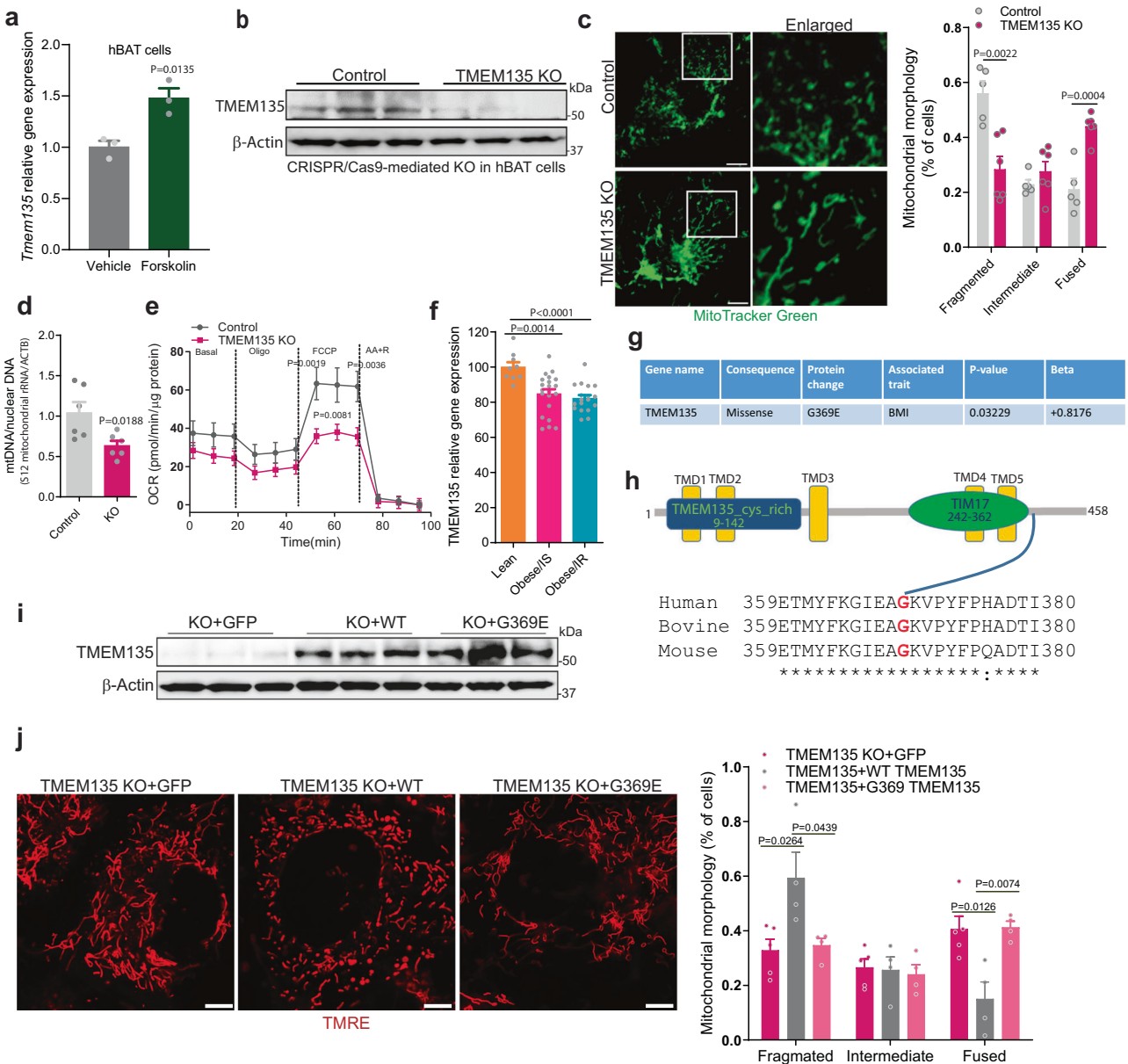

**Fig. 7 | TMEM135 regulates mitochondrial fission in human thermogenic adipocytes. a** qPCR analysis of *Tmem135* expression in human brown adipocytes (hBAT) treated with vehicle (*n* = 3) or 10 μM forskolin (*n* = 3) for 4 h. **b** Western blot analysis of TMEM135 knockout using CRISPR/Cas9 in human brown adipocytes; *n* = 3. **c** Analysis of mitochondrial morphology using confocal microscopy in control (*n* = 5) and TMEM135 KO (*n* = 6) human brown adipocytes stained with Mito-Tracker Green. Plot shows quantification of mitochondrial morphology. Scale bar: 10 μm. **d** mtDNA copy number normalized to nuclear DNA in control (*n* = 6) and TMEM135 KO (*n* = 6) human brown adipocytes. **e** OCR measured in control (*n* = 10) and TMEM135 KO (*n* = 10) human brown adipocytes using a Seahorse XF24 Extracellular Flux Analyzer; oligo oligomycin, FCCP carbonyl cyanide-p-trifluoromethoxyphenylhydrazone, AA + R mixture of antimycin A and rotenone. **f** Relative gene expression of TMEM135 in abdominal subcutaneous WAT of female lean (*n* = 9), obese/insulin sensitive (IS) (*n* = 21), and obese/insulin resistance (IR) (*n* = 18) patients, based on analysis of publicly available transcriptome-profiling data (GEO: GSE94753). **g** A coding variant of TMEM135 associated with increased BMI in a Hispanic population identified in a publicly available AMP-T2D-GENES exome sequencing dataset. **h** Domain architecture and sequence alignment of human, bovine and mouse TMEM135. **i** Western blot analysis of TMEM135 KO human adipocytes transduced with lentivirus expressing GFP (*n* = 3), mouse WT (*n* = 3) or G369E TMEM135 (*n* = 3). **j** Confocal microscopy analysis of mitochondrial morphology of TMEM135 KO human adipocytes transduced with lentivirus expressing GFP (*n* = 5), mouse WT (*n* = 4) or G369E TMEM135 (*n* = 4). Plot shows quantification of mitochondrial morphology. Scale bar: 5 μm. Data are presented as mean ± SEM; statistical significance was determined by two-tailed unpaired Student's *t* test (**a, c, d, f, j**) or 2-way ANOVA with Bonferroni's post hoc test (**e**).

these studies identify TMEM135 as a critical regulator of the thermogenic capacity of brown adipocytes through its ability to control mitochondrial fission.

Brown fat-mediated thermogenesis plays a critical role in the regulation of energy balance. Consistent with its role in promoting thermogenesis, TMEM135 overexpression decreases diet-induced obesity. Conversely, adipose-specific knockout of TMEM135, which impairs thermogenesis, increases obesity in HFD-fed mice. Search of

AMP-T2D-GENES exome sequencing data[48] led to the identification of a coding variant of TMEM135 (G369E) associated with increased BMI. A significant association appears to only exist in a sub-cohort representing 13,000 Hispanic samples[48] and not in the 52,000 overall samples, suggesting that TMEM135 function might be influenced by genetic modifiers. Our studies in human brown adipocytes indicate that this variant is a loss-of-function mutant that impairs mitochondrial fission, suggesting that the increased diet-induced obesity in

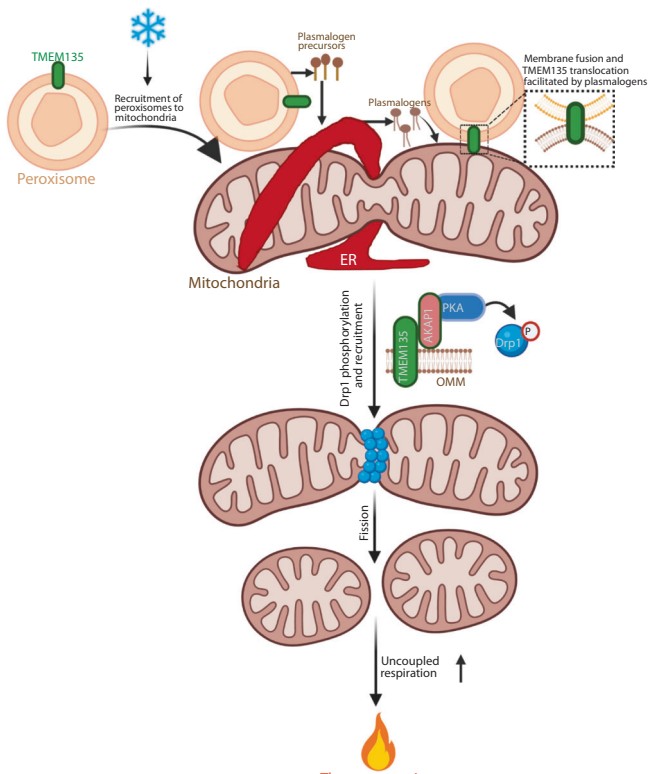

**Fig. 8 | Graphical summary of the peroxisomal regulation of mitochondrial fission and thermogenesis mediated by TMEM135.** Cold promotes recruitment of peroxisomes to mitochondria in brown adipocytes. Peroxisomes cooperate with the endoplasmic reticulum (ER) to synthesize plasmalogens, which are incorporated into mitochondrial membrane. Plasmalogens facilitate membrane fusion and translocation of the peroxisomal transmembrane protein TMEM135 to the outer mitochondrial membrane (OMM). TMEM135 interacts with AKAP1 to promote PKA-mediated phosphorylation of Drp1. This activates mitochondrial fission, leading to increased uncoupled respiration and thermogenesis. Created with BioRender.com.

*Tmem135-AKO* mice is likely related to a defect in brown fat mitochondrial dynamics and thermogenesis. Moreover, adipose-specific knockout of TMEM135 results in insulin resistance, independent of its effect on body weight. TMEM135 overexpression, which promotes glucose tolerance and insulin sensitivity, results in significantly increased RER and a selective upregulation of [18]F-FDG uptake by BAT, suggesting that TMEM135 promotes BAT glucose metabolism to improve systemic glucose homeostasis. These studies are consistent with the notion that the beneficial effects of brown fat extend beyond the regulation of energy balance. In this regard, increasing evidence suggest that BAT also functions as an important metabolic sink for glucose, fatty acids, and branched-chain amino acids and is known to produce batokines, circulating factors that influence insulin sensitivity and metabolic homeostasis[3,63–65]. Our studies identify brown fat TMEM135 as an important regulator of energy balance and whole-body glucose homeostasis that protects against diet-induced obesity and decreases risk for type 2 diabetes.

In summary, our studies reveal a previously uncharacterized organelle interaction regulating mitochondrial fission and energy homeostasis, marked by recruitment of peroxisomes to mitochondria in response to acute β-adrenergic receptor activation. This is followed by peroxisomal lipid-dependent localization of TMEM135 in mitochondria, where it mediates mitochondrial fission by promoting PKA-dependent phosphorylation and activation of Drp1. Beyond providing a mechanistic insight into the role of TMEM135 in mitochondrial fission, our studies identify TMEM135 as a potential target for therapeutic

activation of BAT, perhaps leading to an alternative treatment option for obesity and type 2 diabetes.

## Methods

### Animals

All animal protocols were approved by the Washington University Institutional Animal Care and Use Committee. Transgenic mice that overexpress TMEM135 (*Tmem135[TG]*) under the control of CAG promoter have been previously described[23]. An inbred strain of these mice on the C57BL/6J genetic background was developed and used for all experiments. *Pex16[Lox/Lox]* and *Pex16-AKO* mice have been previously described[20]. *Pex16-AKO* were crossed with *Tmem135[TG]* mice to generate *Pex16-AKO/Tmem135[TG]* mice. *Tmem135[Lox/Lox]* mice on the C57BL/6 J background were generated using the CRISPR/Cas9 system. CRISPR mediated mutagenesis was done by the Genome Engineering and IPSC Center (GEiC) at Washington University. The CRISPR nucleases were validated to ensure that they cut the desired endogenous chromosomal target sites. All successful mutants were verified through deep sequencing. To generate mice with adipose-specific knockout of TMEM135, the floxed mice were crossed with adiponectin-Cre transgenic mice[66]. For each line, respective floxed mice without Cre were used as control. Mice were fed either normal chow (Purina 5053) or a HFD (D12492, Research Diets). Both male and female animals were studied for metabolic phenotyping studies, as indicated in figure legends. All phenotypic data were disaggregated by sex. The animals were kept on a 12 h light/dark cycle and provided ad libitum access to food and water.

For cold or warm challenge studies, mice were individually housed at 4 °C or 30 °C, and body temperature was measured using IPTT-300 transponders. For other analysis related to cold exposure, mice were housed together at 4 °C for 72 hours, unless otherwise noted. For food intake measurements, the mice were singly housed mice and allowed to acclimate for 2 days. Daily food intake was measured by adding known amount of food and subtracting the weight of uneaten food and any crumbs for at least three days.

### Cell lines

Human embryonic kidney 293T (HEK293T) cells were maintained in DMEM supplemented with 10% FBS. Stromal vascular fractions from mouse BAT were isolated, immortalized, and differentiated into adipocytes as previously reported[67]. Briefly, confluent BAT SVF cells were treated with DMEM/F12 supplemented with 0.5 mM iso-butylmethylxanthine, 5 µM dexamethasone, 125 µM indomethacin, 1 µM rosiglitazone, 1 nM T3, and 0.02 µM insulin. After 2 days, the cells were switched to medium supplemented only with 1 nM T3 and 0.02 µM insulin (maintenance medium), which was replaced every 2 days.

Immortalized human brown preadipocytes were established and differentiated as previously described[68,69]. Briefly, differentiation (day 0) was initiated for 10 days using a complete differentiation media (DMEM/F12 medium supplemented with 10% EqualFetal[TM] Bovine Serum [Atlas, #EF-0050-A], 1% antibiotics, 0.5 mM IBMX, 100 nM human insulin, 100 nM dexamethasone, 2 nM triiodothyronine, 10 µg/mL transferrin, 5 µM rosiglitazone, 33 µM biotin, 17 µM pantothenate, and 34 µM BMP7). After induction in the complete media, cells were maintained in DMEM/F12 with insulin (10 nM), dexamethasone (10 nM), biotin (33 µM), and pantothenate (17 µM) until mature adipocytes were used for metabolic experiments (day 20–day 22).

### Plasmid constructs

Lentiviral plasmid encoding shRNA for mouse *Tmem135* (TRCN000258189) was obtained from Sigma-Aldrich (St. Louis, MO). Mito-PAGFP was obtained from Addgene (#23348) and was amplified by PCR using primers Mt-PAGFP NheI Fwd: CGGCGGCTAGCGCCAC CATGTCCGTCCTGACGCCGCT and mt-PAGFP EcoRI Rev: CGCCGGA ATTCTTACTTGTACAGCTCGTCCATG. The resulting amplicon was

cloned into a pLJM1-EGFP plasmid (Addgene #19319) in place of EGFP using restriction sites NheI and EcoRI. TMEM135-BFP was constructed using mTagBFP2-Lifeact-7 (Addgene #54602) and mouse *Tmem135* cDNA (NM_028343) obtained from OriGene. BFP was amplified by PCR using primers NheI BFP2 Fwd: CGGCGGCTAGCGCCACCATGGTGTC TAAGG and BFP2 AgeI Rev: CGCCGACCGGTATTAAGCTTGTGCCC CAGTT. TMEM135 was amplified by PCR using primers TMEM135 AgeI Fwd:   ACACAC   ACCGGTATGGCGGCCCTCAGCAAGTCCAT   and TMEM135 EcoR1 Rev: ACACACGAATTCTCAGGAAAAGTCTATGG GCAG. The TMEM135 resulting amplicon was cloned into a pLJM1-EGFP plasmid (Addgene #19319) in place of EGFP using restriction sites AgeI and EcoRI. In addition, BFP resulting amplicon was cloned into this plasmid in frame with TMEM135 using restriction sites NheI and AgeI. *Tmem135* was knocked out in brown adipocytes by using CRISPR/Cas9 using a previously reported method[70,71]. *Tmem135* sgRNA was designed as TMEM135 Fwd: CACCGACCATCACGACGTTAAGAAA and TMEM135 Rev: AAACTTTCTTAACGTCGTGATGGTC.

## Quantitative real time PCR
Total RNA was isolated using PureLink RNA Mini Kit (Invitrogen 12183018A) and 2 mg total RNA was reverse transcribed into cDNA using the iScriptcDNA Synthesis Kit (Bio-Rad, USA) as previously reported[72]. PowerUp SYBR Green Master Mix was used to conduct quantitative real time PCR. Relative mRNA expression level was determined using the $2^{(-Delta\,Delta\,CT)}$ method and L32 was used as internal reference. Primer sequences are provided in Supplementary Table 1.

## Western blot and immunoprecipitation analysis
Cells or mouse samples were homogenized in RIPA buffer (Cell Signaling Technology, 9806S) or homogenization buffer (0.25 M sucrose, 20 mM HEPES in distilled $H_2O$) containing a protease and phosphatase inhibitor cocktail (Sigma, P8465 and P0001). 40 µg of total protein were subjected to immunoblotting. Proteins were detected with the Odyssey Infrared Imaging System (LI-COR Biosciences). Western blot analysis using TMEM135 antibody from Aviva Systems Biology detected a band at the predicted molecular weight of 52 KD and one slightly above the 37 KD marker, as reported in the vendor's product datasheet. Both bands were decreased in the TMEM135 knockout samples.

Immunoprecipitation assays were performed as described previously[73]. Briefly, cells were homogenized using RIPA Lysis Buffer System (Santa Cruz) supplemented with protease and phosphatase inhibitors. Cell lysates were centrifuged at 13,000 RPM in a microcentrifuge for 10 min to remove unlysed cells. Supernatants were collected and subjected to protein quantification using BCA assay. Cell lysates were then incubated with Anti-FLAG M2 Affinity gel (Sigma-Aldrich, A220) overnight. Immunoprecipitates were washed three times with TBS before elution with SDS-PAGE sample buffer and subjected to SDS-PAGE.

## Purification of recombinant GST-TMEM135 fusion protein
GST-TMEM135 was purified using a modification of our previously described method[74]. Briefly, competent BL21(DE3)pLysS E. coli cells were transformed with the pGEX4T-1 plasmid encoding GST-TMEM135. A 10 ml starter culture was inoculated using a single colony and grown overnight at 37 °C in LB + ampicillin. One day later, the culture was diluted 1:20 into 200 ml of LB + ampicillin. The cells were allowed to grow for 1.5 h at 37 °C, and expression of GST-TMEM135 was induced at 28 °C by addition of 0.1 mM IPTG. Five hour later, bacteria were centrifuged at $4000 \times g$ for 10 min, and the pellet was resuspended in 10 ml of TEN buffer (50 mM Tris-HCl pH 7.4, 0.5 mM EDTA, 0.3 M NaCl). The mixture was vortexed and the following were added: 1 ml of 10 mg/ml lysozyme, 1X CellLytic™ B (Sigma), 0.25 mM DTT, and protease inhibitors. Cells were rocked for 15 min at 4 °C, 0.2% NP-40 was added, and rocking continued for an additional 15 min at 4 °C. Lysates were mixed with 15 ml of a solution containing 1.5 M NaCl/

12 mM $MgCl_2$ then 7 µg/ml DNase was added and the mixture rocked for 2 h at 4 °C. When the suspension was no longer viscous, samples were centrifuged at $15,000 \times g$ for 30 min at 4 °C, the supernatant was transferred to another tube, and then 0.5 ml of a 50% slurry of glutathione agarose beads was added followed by rocking at 4 °C overnight. The lysate was centrifuged at $2500 \times g$ for 5 min, then the beads were washed three times with HNTG (20 mM HEPES [pH 7.5], 150 mM NaCl, 0.1% Triton X-100, 10% glycerol) and two times with PBS/10% glycerol/1 mM DTT. Beads were stored at −20 °C until use in GST pull-down assays.

## GST pull-down assays
GST pull-down assays were performed as previously described[74]. Briefly, cells were collected and lysed using RIPA Lysis Buffer System (Santa Cruz) supplemented with protease and phosphatase inhibitors. Cell lysates were diluted with an equal volume of a detergent-free buffer (10% Glycerol and 1 mM DTT in PBS) containing protease inhibitors. Then, 5 µg of GST alone beads or GST-TMEM135 fusion protein beads were added to the lysates and incubated at 4 °C rocking overnight. Samples were centrifuged at $7000 \times g$ for 2 min and washed four times with TBS before elution with SDS-PAGE sample buffer. The samples were incubated at 95 °C for 5 min, centrifuged at $7000 \times g$ for 2 min, and subjected to SDS-PAGE.

## Mitochondria isolation
Pex16-AKO and control mice were housed together at 4 °C room for 48 h and BAT was harvested. Then, mitochondria were isolated from BAT as previously reported[75]. Briefly, BAT depots were collected from Pex16-AKO and control mice after perfusion using PBS. The tissues were cut into small pieces and homogenized in 100 µl ice cold homogenization buffer (0.25 M sucrose, 20 mM HEPES in distilled $H_2O$) containing a protease and phosphatase inhibitor cocktail (Sigma, P8465 and P0001) for 10 strokes in a dounce homogenizer. The pestle was washed with 800 µl of homogenization buffer to remove any residual homogenate. The homogenates were transferred to 1.5 ml tubes and vortexed thoroughly before centrifugation at $800 \times g$ for 10 min at 4 °C. The upper fat cake was removed by forceps and the supernatant (~800 µl) was transferred to a new tube and centrifuged at $10,000 \times g$ for 10 min at 4 °C. The supernatant was discarded and the pellet (mitochondrial fraction) was washed one time and then used immediately for mass spectrometry-based proteomics or stored at −80 °C for future use.

## Mitochondrial proteomics
Mass spectrometry-based analysis of the mitochondrial proteome was done by the Proteomics Shared Resource at Washington University. Briefly, mitochondrial isolates were digested to generate peptides, which were labeled with tandem mass tag reagents (TMT-11) and analyzed by liquid chromatography coupled to mass spectrometry (LC-MS). Mitochondrial pellets were solubilized with 100 µl of SDS buffer (4% (wt/vol), 100 mM Tris-HCl pH 8.0). Protein concentration determined using a Pierce™ BCA Protein Assay Kit. Protein disulfide bonds were reduced using 100 mM DTT with heating to 95 °C for 10 min. Peptides were prepared as previously described using the filter-aided sample preparation method as previously described[76]. A reference pool was made from lysate of all 10 samples in the study and processed alongside the samples. The samples (30 µg in a normalized volume of 30 µL SDT buffer) were mixed with 200 µl of 100 mM Tris-HCL buffer, pH 8.5 containing 8 M urea (UA buffer). The samples were transferred to the top chamber of a 30,000 MWCO cutoff filtration unit (Millipore, part# MRCF0R030) and processed to peptides as previously described[77]. The peptides were dried in a Speedvac concentrator (Thermo Scientific, Savant DNA 120 Speedvac Concentrator) for 15 min. The dried peptides were dissolved in 1% (vol/vol) trifluoroacetic acid (TFA) and desalted using two micro-tips (porous

graphite carbon, BIOMEKNT3CAR) (Glygen) on a Beckman robot (Biomek NX), as previously described[78]. The peptides were eluted with 60 μl of 60% (vol/vol) acetonitrile (MeCN) in 0.1% (vol/vol) TFA and dried in a Speed-Vac (Thermo Scientific, Model No. Savant DNA 120 concentrator). The peptides were dissolved in 20 μl of 1% (vol/vol) MeCN in water. An aliquot (10%) was removed for quantification using the Pierce Quantitative Fluorometric Peptide Assay kit (Thermo Scientific, Cat. No. 23290). The remaining peptides were lyophilized in preparation for labeling with isobaric mass tags. Dried peptides were dissolved in 40 μl of HEPES buffer (100 mM, pH 8.5) and 40% by volume were labeled according to the vendor protocol using the TMT-11 reagent kit. The labeled peptides were combined and desalted using micro-tips. Peptides were eluted in 60 μl of 60% (vol/vol) MeCN in 0.1% (vol/vol) TFA, transferred to autosampler vials and dried in a Speed-Vac (Thermo Scientific, Model No. Savant DNA 120 concentrator) and stored at −80 °C.

The samples were analyzed using ultra-high performance mass spectrometry[79] using a hybrid quadrupole Orbitrap LC-MS System, Q-Exactive™ PLUS interfaced to an EASY-nano-LC 1000). A 75 μm i.d. × 50 cm Acclaim PepMap 100 C18 RSLC column (Thermo Scientific™) was equilibrated with 100% solvent A (1% formic acid) on the nano-LC for a total of 11 μl at 700 bar pressure. Samples in FA 1% (vol/vol) were loaded at a constant pressure of 700 bar. Peptide chromatography was initiated with mobile phase A (1% FA) containing 2% solvent B (100% ACN, 1%FA) for 5 min, then increased to 20% B over 100 min, to 32% B over 20 min, to 95% B over 1 min and held at 95% B for 19 min, with a flow rate of 250 nl/min. Data were acquired in data-dependent mode. Full-scan mass spectra were acquired with the Orbitrap mass analyzer using a scan range of $m/z = 325$ to 1800 and a mass resolving power set to 70,000. Ten data-dependent high-energy collisional dissociations were performed with a mass resolving power at 17,500, a fixed lower value of $m/z$ 110, an isolation width of 2 Da, and a normalized collision energy setting of 27. The maximum injection time was 60 ms for parent-ion analysis and product-ion analysis. Ions that were selected for MS/MS were dynamically excluded for 30 s. The automatic gain control was set at a target value of 1e6 ions for full MS scans and 1e5 ions for MS2.

## Subcellular fractionation of brown adipocytes
Peroxisomes were isolated from cultured brown adipocytes using Peroxisome Isolation Kit (Sigma-Aldrich) with modifications of the manufacturer's protocol. All reagents were prechilled on ice prior to the start of the assay and all steps were performed on ice to preserve organelle integrity. Briefly, $4 \times 10^7$ brown adipocytes were washed with PBS and resuspended in 1× Peroxisome Extraction Buffer (5 mM MOPS, pH 7.65; 0.25 mM sucrose, 1 mM EDTA, and 0.1% ethanol). The cells were broken using a dounce homogenizer and centrifuged at $1000 \times g$ for 10 min to collect the nuclear pellet. The supernatant was then transferred to a new tube and centrifuged at $10,000 \times g$ for 10 min to collect the mitochondria-enriched pellet. The supernatant was transferred to a new tube and centrifuged at $21,000 \times g$ for 40 min to collect the crude peroxisome fraction (CPF). The supernatant was transferred to a new tube as the cytosol fraction. To isolate the peroxisome fraction, CPF was further fractionated using OptiPrep™ density gradient medium and centrifugation at $100,000 \times g$ for 1.5 h, as described in the manufacturer's protocol. The nucleus, cytosol, mitochondria, and peroxisome fractions were subjected to SDS-PAGE and immunoblotted using organelle-specific antibodies.

## Indirect calorimetry
To measure basal $VO_2$, carbon dioxide production ($VCO_2$), and respiratory exchange ratio (RER) in HFD-fed animals, a PhenoMaster (TSE Systems) metabolic cage system was used. Mice were acclimated to the system for three days before measurements were taken. To measure CL316,243-stimulated $VO_2$ in mice, the mice were acclimated

to the system for 4 h and measurements were done for 1 h prior to and 3 h after administering CL316,243 (1 mg/kg, i.p. injection).

## Home-cage $CO_2$ measurements
$CO_2$ was measured with a nondispersive infrared (NDIR) $CO_2$ sensor (Sensirion SCD30), sampling at 1 sample/minute. This $CO_2$ sensor was connected to a wireless MCCI Catena Node (MCCI Corp) which also measured humidity, temperature, activity. The device was enclosed in a 3D printed housing and mounted inside of a regular rodent home-cage. Ambient in-cage $CO_2$, temperature, humidity, and activity levels were wirelessly transmitted from the cage to a cloud database for visualization and analysis. This system is battery powered and portable, which was necessary for taking these measurements in a warm room that was kept 30 °C.

## mtDNA copy number
Total genomic DNA was isolated from brown fat tissue or BAT SVF cells using a DNeasy Blood & Tissue Kit (QIAGEN) as previously reported[20]. mtDNA copy number normalized to nuclear DNA was measured by qPCR using 50 ng total genomic DNA and PowerUp SYBR Green reagent. Primer sequences for mitochondrial and nuclear DNA were as follows: mitoDNA-Fwd, TTAAGACACCTTGCCTAGCCACAC; mitoDNA-Rev, CGGTGGCTGGCACGAAATT; NucDNA-Fwd, ATGACGATATCGCT GCGCTG; NucDNA-Rev, TCACTTACCTGGTGCCTAGGGC.

## Immunofluorescence and confocal microscopic imaging
Frozen sections or cells samples were fixed with ethanol or 4% paraformaldehyde, followed by primary antibody and the corresponding secondary antibody incubation. The slides were imaged using a Nikon A1Rsi Confocal Microscope. The fluorescent intensity and colocalization were calculated using ImageJ.

## Measurement of OCR in cultured adipocytes
Mitochondrial respiration was determined using the Seahorse XFe24 Extracellular Flux Analyzer and the XF Cell Mito Stress Test Kit according to the manufacturer's instruction (Agilent Technologies, Santa Clara, CA). Briefly, BAT SVF cells were seeded in a XF24 cell culture microplate before treatment with differentiation media, as previously described[20]. On day 4, the cells were treated with scrambled, TMEM135 shRNA, pLJM1-EGFP or TMEM135 overexpression lentivirus in the maintenance media. After 48 h of treatment, the viral media were removed and replaced with fresh maintenance media. On day 9 of differentiation, cells were washed with XF Assay Media, pre-incubated in a non-$CO_2$ incubator at 37 °C for 1 h in XF Assay Media. For the assay, cells were treated sequentially with final concentration 1 μM oligomycin, 1.5 μM FCCP (carbonyl cyanide-p-trifluoromethoxyphenylhydrazone), and 1 μM rotenone plus 1 μM antimycin A. The Seahorse Wave (RRID: SCR_014526) software was used to run and collect the results for all XF assays. All reagents were purchased from Agilent Technologies.

## Mitochondrial respiration measurement in frozen BAT tissues
Frozen BAT tissue mitochondrial respiration was measured as previously described[80]. Briefly, frozen BAT tissues were thawed in ice-cold PBS, minced, and homogenized in MAS buffer (70 mM sucrose, 220 mM mannitol, 5 mM $KH_2PO_4$, 5 mM MgCl2, 1 mM EGTA, 2 mM HEPES pH 7.4). BAT samples were mechanically homogenized with 10−20 strokes. All homogenates were centrifuged at $1000 \times g$ for 10 min at 4 °C; then, the supernatant was collected. Protein concentration was determined by BCA (ThermoFisher). 3 μg BAT tissue protein was loaded in XF24 cell culture microplate plate. Substrate injection was as follows: 5 mM succinate + rotenone (5 mM + 2 μM) were injected at port A; rotenone + antimycin A (2 μM + 4 μM) at port B; TMPD + ascorbic acid (0.5 mM + 1 mM) at port C; and azide (50 mM) at port D.

## Mitochondrial morphology assay

BAT SVF cells were seeded in 24-well plates with high performance #1.5 cover glass bottoms (Cellvis P24-1.5H-N), grown to confluence and then treated with brown adipocyte differentiation cocktail as previously described[20]. Four days after the initiation of differentiation, the cells were transduced with scrambled, TMEM135 shRNA, pLJM1-EGFP or TMEM135 overexpression lentivirus in the differentiation medium, as indicated. After 48 h, the viral medium was removed and fresh differentiation medium was added. On day 9 of differentiation, the medium was removed and the cells were washed with pre-warmed PBS before addition of phenol red-free DMEM containing tetramethylrhodamine, ethyl ester (TMRE) to label mitochondrial networks, and the cells were imaged using a Zeiss LSM 880 Airyscan Two-Photon Confocal Microscope. Mitochondrial morphology was quantified by visual inspection, and cells were divided into three groups, exhibiting fragmented, fused, or intermediate (partially fragmented) mitochondria, as previously described.

## Mitochondrial fusion assay

In this assay, mitochondria in cells expressing mtPA-GFP are fluorescently activated and their subsequent fusion is measured by fluorescent quantification[24]. Briefly, BAT SVF cells were isolated from control and Pex16-AKO mice and then were immortalized. These cells were seeded in glass-bottom plates (Cellvis #P24-1.5H-N), infected with lentiviral mtPA-GFP and differentiated into adipocytes as previously described[24]. To visualize the entire mitochondrial network, the cells were stained with TMRE. Individual mitochondria were photoconverted by laser pulse using a Zeiss LSM 880 Airyscan Confocal/Two-Photon Microscope. The diffusion of mtPAGFP was assessed by continuously imaging at 15 min intervals.

## Histology

IWAT, gWAT, BAT and liver tissues were fixed overnight in 10% neutral buffered formalin solution before being dehydrated and embedded in paraffin. Deparaffinized 5 μm sections were stained with hematoxylin and eosin. Slides were scanned using microscopy.

## Transmission electron microscopy

TEM analysis of BAT was performed as previously reported[20]. Briefly, mice were sacrificed and perfused with warm Ringer's solution (Fisher Science) for 2 min at 37 °C, followed by 2% paraformaldehyde plus 2.5% glutaraldehyde, in 0.09 M cacodylate buffer, pH 7.2, containing 5% sucrose and 0.025% $CaCl_2$ at 37 °C for 5 min. BAT was gently removed without pulling or stretching and placed into generous amount of room temperature fixative. The tissue was cut into smaller pieces (10 × 10 mm) while keeping it immersed in the fixative. The tissue fixation was continued overnight at 4 °C using a shaker/rotator. Images were acquired using a JEOL JEM-1400 Plus Transmission Electron Microscope.

## ELISAs

The following kits were used to measure serum factors: insulin and adiponectin. The ELISA kits were purchased from Crystal Chem (Downers Grove, IL). Manufacturer's recommended protocols were used for all serum measurements.

## Insulin signaling assay

To measure insulin signaling in the liver, mice were fasted for 4–6 h before tissue was collected and snap frozen in liquid nitrogen at the basal state or 10 min after an IP injection of 5mU/g of insulin (Humulin R, Lilly). Tissue was placed in tissue lysis buffer (0.1% SDS, 0.1% sodium deoxycholate,1% Triton X-100, 50 mM Tris-HCL pH 7.6, 5 mM EDTA, 150 mM NaCl, 5 mM NaF, and protease inhibitor) and homogenized (Glas-Col099C-K54).

## Live imaging of TMEM135 in mitochondria and peroxisomes

BAT SVF cells expressing TMEM135-BFP were seeded in glass bottom dish (Cellvis D35-20-1.5-N), grown to confluence and then treated with brown adipocyte differentiation cocktail as previously described[1]. Then this stable cell line was transduced with GFP-SKL lentiviral particles[72]. On day 8 of differentiation, the medium was removed and the cells were washed with pre-warmed PBS before addition of phenol red-free DMEM containing TMR) to label mitochondrial networks, and the cells were imaged using a Zeiss LSM 880 Airyscan Two-Photon Confocal Microscope.

## In vivo metabolic imaging with [18]FDG-PET/CT

$TMEM135^{TG}$ and WT mice fed a HFD were administered a dose of -150 μCi of [18]FDG via a tail-vein injection under 2% isoflurane anesthesia. Mice received CL316,243 at dose of 1 mg/kg of bodyweight 30 min before preclinical PET/CT imaging as previously reported[81]. Whole mouse imaging was performed on the Mediso nanoScan PET/CT imaging system at the Preclinical PET Imaging Facility at Washington University in St. Louis. Subsequently, mice were euthanized, and their interscapular BAT, inguinal WAT, epididymal WAT, liver and skeletal muscle were collected for measurement of radioactivity represented as percent injected dose per gram of tissue. [18]FDG images were quantified by determining the standardized uptake value (SUV).

## Quantification and statistical analysis

Data are expressed as mean ± SEM unless stated otherwise. Comparisons between two groups were performed using an unpaired, two-tailed t-test. ANOVA was used for more than two groups. A P-value less than 0.05 was considered significant. Statistical analysis and graphs were generated using GraphPad Prism software. In imaging experiments, the number of independent experiments is indicated in figure legends and if the cells were counter-stained with DAPI, the number of cells used in quantification is also indicated. In selected imaging experiments (when indicated in figure legends), quantification was performed with the investigator blinded to the identity of the samples.

## Reporting summary

Further information on research design is available in the Nature Portfolio Reporting Summary linked to this article.

## Data availability

The mass spectrometry proteomics data presented in Fig. 2 have been deposited in the Mass Spectrometry Interactive Virtual Environment (MassIVE) repository and are publicly available under accession code MSV000090693. All other data supporting this study are available within this Article and Supplemental Information. Source data are provided with this paper.

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

## Acknowledgements

This work was supported by NIH grants DK132239, DK118333, DK115867, and EY022086. The core services of the Washington University Diabetes Research Center (DK020579) and the Nutrition Obesity Research Center (DK056341) also provided support for this work. DH was supported by a Diabetes Research Postdoctoral Training Program fellowship (T32 DK007120). Confocal microscopy and transmission electron microscopy were performed through the use of Washington University Center for Cellular Imaging (WUCCI), supported by Washington University School of Medicine, The Children's Discovery Institute of Washington University and St. Louis Children's Hospital (CDI-CORE-2015-505 and CDI-CORE-2019-813), and the Foundation for Barnes-Jewish Hospital (3770 and 4642). The proteomic experiments were performed at the Washington University Proteomics Shared Resource (WU-PSR). The WU-PSR is supported in part by the WU Institute of Clinical and Translational Sciences (NCATS UL1 TR000448), the Mass Spectrometry Research Resource (NIGMS P41 GM103422; R24GM136766) and the Siteman Comprehensive Cancer Center Support Grant (NCI P30 CA091842). The authors thank Dr. Jason Flannick (Broad Institute, Harvard Medical School) for reading the manuscript and providing helpful comments related to TMEM135 variants.

## Author contributions

D.H. designed and conducted experiments and performed data analysis. M.T., D.L., X.L., and H.P. performed experiments. B.K and D.L. performed data analysis. A.V.K and K.I.S. designed experiments and performed data analysis. Y-H.T., B.R., and A.I. designed experiments and provided reagents. I.J.L. conceived the study, designed experiments, and performed data analysis. D.H. and I.J.L. wrote the paper. All authors read the manuscript and provided comments.

## Competing interests

The authors declare no competing interests.
