## [Peer Review File · Nature Communications]

TMEM135 Links Peroxisomes to the Regulation of Brown Fat Mitochondrial Fission and Energy HomeostasisREVIEWER COMMENTS

Reviewer #1 (Remarks to the Author):

Hu and colleagues explore the impact of deficient peroxisome biogenesis (via the deletion of the Pex16 gene) on mitochondria morphology. Through a series of experiments, the authors demonstrate that Pex16 deletion leads to a highly fused mitochondrial network and postulate that this is due to altered synthesis of glycerophospholipids. This prevents the translocation of TMEM135 from the peroxisomes to the mitochondria, which in turn is necessary to engage the mitochondrial fission process.

The topic and concepts presented are really interesting. Through the manuscript, the authors use an extensive combination of primary cultures, genetically engineered mouse models and molecular biology analyses. The manuscript is well written and easy to read. Nevertheless, there are some aspects that deserve attention. The work often spreads too thin and some aspects remain a bit preliminary. Also, the manuscript lacks consistency in some of its approaches and the description of the experiments and methods could be more precise. Some suggestions to improve the manuscript can be found below:

1/ In Figure 1C, the authors argue that Pex16 deficient adipocytes show compromised fission. This, however, is difficult to evaluate. Even if fission occurred, the dye will not dilute unless there are other fusion events. Rather, the fact that the green fluorescence also argues for impaired fusion. The difficulty to evaluate this aspect also comes from the fact that changes in the amount/intensity of fluorescence are, at least to me, not possible to judge from the images. Some quantifications could help.

2/ The authors repeatedly observe very drastic changes in mtDNA content upon alterations of Pex16, GNPAT or TMEM135 levels. This aspect is extremely important for the phenotypes observed, yet a clear explanation is absent. The authors speculate that this could be related to deficient fission, yet never really demonstrate this point. It would be important to, at least, demonstrate that there is impaired mtDNA replication and explore potential alterations of players involved in DNA replication processes, such as TFAM. The authors might also want to discuss this finding in the perspective of other models of impaired fission or enhanced fusion. Finally, it would be beneficial to confirm such drastic changes by using two additional targets for their mtDNA vs. nDNA measurements.

3/ Some measurements were done, out of the blue, on cold-exposed mice rather than in mice under normal housing conditions (e.g.: Extended Data Fig.1, proteomic analyses in Fig.2, Fig.4F). The authors do not provide a clear rationale for this choice and the results might highly be biased by the treatment. Could the authors do some test to corroborate whether the observation is also occurring in mice at normal housing temperature?

4/ For the proteomic analyses on mitochondrial fractions, the authors should provide some analyses to certify the purity (or clear enrichment) of the mitochondrial fraction.

5/ The authors argue that Pex14 and Phyh have mitochondrial presence in Pex16 KO mice. However, this is not proven. It could be either marginal/immature peroxisomal content or contamination of other fractions. Can the authors support this statement with some imaging data?

6/ The authors should evaluate the localization of TMEM135 through cellular fractionation techniques in order to better estimate the fraction of the total TMEM135 content that resides in the mitochondria.

7/ In most imaging experiments the TMEM135 signal is difficult to interpret and the experiments would definitely benefit from some quantifications. A second aspect that could be improved is that large areas of these cells are lipid droplets. Maybe not in all experiments, but at least in the most critical ones, visualization of lipid droplets could provide a more accurate view of the cellular organelle.

distribution.

8/ Does FCCP induce mitochondrial fission in TMEM135 KD cells?

9/ In Figure 3H, the AA+R value should be subtracted all along the exp. Based on the results presented, the differences in respiration might be blunted once corrected by AA+R values. Second, the decrease in respiration in TMEM135 KD cells upon FCCP treatment suggests that mitochondria might be more sensitive to it. Could the experiment be repeated with a lower FCCP concentration?

10/ Figure 4G there is a clear sex dimorphism in the influence of cold exposure on core body temperature. Is this dimorphism also observed in mitochondrial architecture or mtDNA content?

11/ At least for me, it is difficult to make any conclusion from Fig.6A. The mitochondrial architecture in these images is impossible to visualize. Peroxisomes seem as big as mitochondria, and some look like actually exactly the same detection element. This problem extends all along figure 6 and Extended Data Figure 6, where all imaging studies should be quantified.

12/ Given the relation of TMEM135 on thermogenesis, could the authors evaluate if its knock-down or overexpression alters adrenergic signaling?

Minor:

1/ Figure 1a – the experimental design should be clarified. Are these adipocytes from Pex16 floxed mice that are infected with the Cre recombinase or are they from Pex16 floxed mice (control) and Pex16 AKO mice?

2/ Figure 1e - the enzymes names should be spelled out in the legend

3/ Figure 1f/g - data for GNPAT protein levels should be provided

4/ Extended Data Figure 2D: the images presented are not really convincing. First total homogenates should be run next to the different fractions in order to appreciate enrichments. Second, some bands look really weird and don't really align. Finally, some of the outcomes are unclear (Akt presence in the mitochondria). I would strongly encourage the authors to repeat the experiment taking into account the points above.

5/ Extended Data Figure 4b: why is there such little uncoupling in these mitochondria? Could this be linked to the fact that Complex II, instead of Complex I is being tested?

6/ Extended Data Figure 4m: could the authors please provide absolute blood glucose values?

7/ Figure 5E: This result is surprising, as thermoneutrality generally takes longer (around 4 weeks) to have such a strong impact on BAT-related thermogenesis. Could the authors please discuss how come the wild-type mice are so impaired and clarify whether these mice were males or females?

8/ Figure 5F: when compared to the TMEM135 KO mice, it seems that it is the controls who actually change, being much heavier in this intervention. Can the authors at least discuss that?

9/ Extended Data Fig.5E: O₂ and VCO₂ values should be corrected by lean mass.

10/ Extended Data Fig.5H: This is a very preliminary experiment, which departs quite significantly from the techniques used in this or other papers to evaluate metabolic flexibility. Proving this point would take many more measurements, and therefore, I would instead suggest to remove these two

panels.

11/ Figure 5M: why is the Akt band lower in insulin-treated mice? In general, insulin treatment should lead to a slightly higher band.

12/ Extended Data Figure 5I: is this insulin stimulated or basal state? Also, shouldn't the results in BAT be corrected by the fact that it has a lower lipid content per gram of tissue?

13/ How would these markers look in the basal state?

Reviewer #2 (Remarks to the Author):

Mitochondria are dynamic organelles that undergo fission and fusion. Fission and fusion of mitochondria control their metabolic activity, including with regards to their thermogenic capacity in brown adipocytes. The metabolic activity of peroxisomes also influences the thermogenic activity of brown adipocytes. Here, Hu and coauthors present evidence that the peroxisomal membrane protein TMEM135 is a regulator of mitochondrial fission and thermogenesis. Thermogenic stimuli promote association between peroxisomes and mitochondria and plasmalogen (specialized lipids made in part by peroxisomes)-dependent localization of TMEM135 to mitochondria. Knock out of TMEM135 in brown adipose tissue in mice blocks mitochondrial fission, impairs thermogenesis and increases diet-induced obesity and insulin resistance, while overexpression of TMEM135 counteracts these effects. At mitochondria, TMEM135 is involved in the retention of the fission factor, Drp1.

This manuscript describes an interesting communication pathway between peroxisomes and mitochondria mediated by the movement of the membrane protein TMEM135 from the peroxisome to the mitochondrion in brown adipocytes to regulate the thermogenic activity of brown adipose tissue. The findings presented in this manuscript will represent an important contribution to our understanding of the control of the metabolic activity of brown adipose tissue and the role played in this process by communication between peroxisomes and mitochondria once the authors have addressed a number of critical issues as outlined below:

Principal Issues:

1) The fluorescence microscope images purporting to show fusion or fission of mitochondria have to be dramatically improved. While an image such as Figure 1b clearly demonstrates fusion of mitochondria in Pex16 KO cells versus control cells, most fluorescence images do not convincingly show either fusion or fission of mitochondria as the authors conclude one way or the other throughout the manuscript. Since these microscope images form the foundation of most of the conclusions of this manuscript, they must be dramatically improved to convincingly support the conclusions of the authors.

2) Scale bars must be added to all micrographs. Given that images presented in figures are of many different sizes, simply stating that original images were taken, for example, at 60× magnification is not particularly helpful.

3) The authors should present fluorescence microscopy images of mitochondria corresponding to the electron microscopy images of Figure 4d and Extended Data Fig. 6e.

- 4) Figure 1c. Why is there green mitochondrially targeted photoactivatable GFP (mtPA-GFP) seen in the PEX16KO cells? If indeed the mtPA-GFP is targeted to mitochondria it should present exclusively as yellow due to colocalization with the mitochondrial marker, TMRE. Please explain.
- 5) Why is there a TMEM135 band (of the same size as found in the Control) in the western blot of the brown adipose tissue coming from the TMEM135-AKO mouse?
- 6) A figure showing the structure of the entire mouse TMEM135 gene would help guide the reader.
- 7) The migrations of molecular weight standards must be added to every western blot.
- 8) Extended Data Fig. 6a. Pex16-KD cells, not Pex16-KO cells.
- 9) Figures 6e and 6f. TMEM135OE not simply TMEM135.
- 10) Figure 6f showing recruitment of Drp1 to mitochondria by overexpression of TMEM135 is unconvincing. New images are needed, along with quantification.

Minor Issues:

- 1) p. 7, line 174. ‘...five predicted transmembrane domains...’.
- 2) p. 8, line 197. ‘Extended Data Fig. 3a’ NOT ‘Extended Data Fig. 3e’.
- 3) p. 8, line 197. For the non-expert, what does the addition of norepinephrine do to brown adipocytes?

Reviewer #3 (Remarks to the Author):

Nature communications reviews:

TMEM135 Links Peroxisomes to the Regulation of Brown Fat Mitochondrial Fission and Energy Homeostasis

Hu et al describe a role for the protein TMEM135 in the regulation of brown fat mitochondrial fission and, as a result, thermogenesis in mice. They propose that this protein provides a link between peroxisomes and mitochondrial – peroxisomes are known to regulate mitochondria morphology via specific lipid species. Hu et al. propose that TMEM135 forms part of this pathway. The Authors provide strong evidence that loss of TMEM135 in adipose tissue in mouse increased susceptibility to obesity and glucose intolerance on a HFD and decreased cold tolerance. The authors propose a model whereby TMEM135 acts a link between peroxisomes and mitochondria by physically translocating to mitochondria to help regulate mitochondrial fission via DRP1.

The manuscript is generally clearly written, but is quite wordy and long in some sections. Also, the Authors use phrases like “remarkably” quite frequently, which in most cases is not needed (obviously this is a personal preference).

The data showing a key role for TMEM135 in brown fat function and energy metabolism is convincing, in general. I have more reservations over the molecular details provided in the manuscript, largely due to 1) a real lack of methodological details, 2) a lack of orthologous approaches to show TMEM135

translocation to mitochondria, for example, 3) a lack of unbiased image analysis and 4) representative images that are generally highly overexposed and also do not obviously match the authors interpretation.

I have split my review into two sections – in vivo data and cell/molecular data. Note that my expertise falls more on the side of the cell/molecular data.

In vivo

The increase in TMTM135 with cold is stark. Does this match other peroxisome proteins, or is this unique to TMEM135? In addition, the protein increase looks much more than the mRNA – is this correct? Is this of interest?

Cold stress test data –the data within the paper appear to be inconstant. In Fig. 5, the WT mice decline in temperate across 6 h, reaching 30 oC after 6 hours. In Fig. 4G and EDFig. 4C, control mice plateauing at 34oC after 1-2 h cold. Are data from this assay usually this variable? Is the difference due to how the mice were housed prior to the cold test?

For GTT and ITT data: GTT and ITT data are typically analysed using an area under/over the curve to describe the entire response to glucose/insulin over the test. However, the analyses are a little inconsistent here with AUC shown for the ITT and not GTT. I would suggest that AUC should be provided for GTT (corrected for 0 min) and area over the curve for ITT.

In addition, presenting data from ITT as % of initial glucose can be misleading, as it doesn't show the difference in initial blood glucose. Please see PMID: 34117483. In general, the blood glucose at 0 min GTTs shows very little difference in blood glucose, so I would expect this % analysis to be unnecessary (assuming the same length fast). Please plot raw data for ITTs.

In Fig. 4M – The Authors interpret these data as evidence for insulin resistance. The increase in blood glucose at 15 min is likely a stress response, as an increase with insulin is not expected. This is the only time point that shows a difference, so its not very clear there is an insulin resistant phenotype in these AKO mice from this test. Perhaps an AOC analysis would provide more insight as the control mice have a generally lower glucose over the course of the experiment. Whether or not these mice are insulin resistant as measured by ITT is not critical to the paper, but this needs to be interpreted correctly.

How was the NE, insulin and glucose data calculated. Was this on BW or lean mass? Could the changes in BW be a factor in improved glucose tolerance, for example, if dosed on BW?

How was food intake measured? Was this measured as a group, or in singly housed mice? How long was food intake measured for?

Molecular/Cell:

Mitochondrial proteomics:

The details on how mitochondria were isolated before proteomics is not provided in any detail, and, as far as I can see, are not provided in the reference the Authors point us towards for a detailed method (REF 19). Please include the experimental details here. In general, the proteomics data re not described in detail, and more detail could be provided. For example, "Many proteins either increased or decreased as a result of peroxisome deficiency" – the authors should provide exact numbers here. Do other peroxisomal proteins go down in the proteomics data? The authors suggest that some peroxisomal proteins are increased in mitochondria in peroxisome deficit cells due to "mistargeting" to mitochondria. Do the Authors have evidence that these proteins now localise to mitochondria by an orthologous method? Are these the only peroxisome proteins that went up – is TMEM135 the only peroxisome protein that went down?

Organelle preps

There is no detail on how the organelle preps in ED Fig 2D were performed. Please provide details. This information is critical for interpreting these data. For example, are ER proteins present in the Nuc fraction? Could it be that TMEM135 is in the ER, not the Nucleus? Also, ref 19 has a similar blot but this blot has obvious peroxisomal contamination in the mitochondrial fraction. Here, the preps look very clean, despite it being very challenging to separate mitochondria and peroxisome by differential centrifugation. Nevertheless, it does look like TMEM135 is more multilocalised between mitochondria and peroxisomes than PMP70 – what about other peroxisome proteins? In addition, contrast and banding on this blot are strange – for example AKT bands do not align with the label. Would an organelle IP approach provide a purer method to separate mitochondria and peroxisome proteins?

Imaging data – image quality, quantification and representative images:

In general, the standard of the microscopy images makes them hard to interpret. There are often over-exposed to that the mitochondrial signal is very strong and it's hard to see individual structures. The Authors use the term "merged" for colocalisation between green and red signals, I think. This is not the correct terminology. Also, separate images should be shown. This is critical

Also, the interpretation provided by the Authors in some cases is not clearly matched by the representative images. For example – Figure 7J. I could not pick which of the images corresponded with the quantitation from viewing these representative images.

This would be aided by 1) quantifying all images in an unbiased way and/or 2) blinding the analysis of mitochondrial fragmentation if this isn't how this was done already.

Fig. 2F – The Mander's overlap coefficient. Why was this method chosen to determine colocalisation? Where analysis like this has taken place, how many cells do the 3 ns in the graph represent? This information should be provided in all cases?

Fig 1C – why is there a green section – this could colocalise with the TMRE stain? Can C be quantified? 1GH – the image looks very blurry. Can the authors provide a better one?

Figure 3E - Were investigators blinded for the analysis?

Figure 4D – the EM images show increased density around mitochondria – is this a consistent feature? What does this mean?

Figure 4F – 8 cells per condition seems low. How many different mice was this from?

Figure 6a – TOMM20 is a mitochondrial marker, so why does this look so different from the TMRE staining in other images throughout the paper?

Mitophagy data – MitoKeima – ED Fig. 1B. Can this be quantified?

Fig 1G – this needs to be quantified in some way – there are ways to quantify the Mito network.

Imaging data – mitochondria-peroxisomes contacts:

The data showing apparent increase in peroxisome-mitochondrial contacts with NE are not overly convincing. 6b – the GFP-PTS-1 signal doesn't seem to change much with NE, but the mitochondrial morphology does. Could the increase in apparent signal overlap be technical, and due to the change in what the mitochondrial signal looks like, rather than genuine contacts? – here the mito signal is very highly exposed as well, making it hard to discern if the green and red are overlapping or just in close proximity.

Overall, these imaging data need to be paired with orthogonal assays (biochemistry?) or a more extensive imaging analysis (e.g. EM, high resolution) to really probe the interactions between peroxisomes and mitochondria.

Imaging data – TMEM135 in mitochondria:

The authors conclude, from these images in Fig. 2F, that mitochondrial localisation of TMEM135 was lower in PEX16 KO cells – but the images do not convincingly show that TMEM135 is in mitochondria. If TMEM135 was in mitochondria, as the authors suggest, I would expect it to begin to look like mitochondria in morphology, not remain as signal puncta near mitochondria. It maybe that this is the case, but separate images are not shown. Data from REF 22 (elife) also not showing mitochondrial localisation – but more like the association of TMEM135 vesicles/puncta close to mitochondria. In PEX16 KO cells do not have peroxisomes. So where is the TMEM135 localised? The TMEM135 is still

in puncta, but this cannot be peroxisomes. Fig. 2F – there are a large number of TMEM135 puncta very distant from mitochondrial stain in the Pex16 KO image. Is this image representative? In all other images the whole cells are full of mitochondria. Why is there a large gap in this image. Is it the nucleus?

In 6c – the BFP signal looks to be less punctate and more dispersed – it is the dispersed signal that overlaps with the mitochondria – giving the large increase in the overlap coefficient. Or is this picking up small contacts between puncta (presumably peroxisomes) and mitochondria. It would be really useful to see another peroxisome marker here – is this dispersed signal specific to TMEM135?

The claimed increase in TMEM135 in mitochondria with NE shown in 6C would be more convincing if paired with orthologous methods, such as blotting for TMEM in mitochondrial fractions. The same is true for other data in the paper, such as the in Fig 6F, which does not convincingly show more DRP in mitochondria with TMEM135 OE.

Interactions data (PD, IP): How were Ips and pull downs performed. What buffer was used? This is important for how we interpret these data. For example in EDFig2C – Can the Authors confirm which buffer was used for the immunoprecipitation? The methods are very unclear on this. If a sucrose buffer was used, then the interaction between TMEM135 and PEX19 is not convincing as the IP may just be pulling down whole peroxisomes. Similarly, is the flag-tag of TMEM135 cytosolic, or luminal?

TMEM135 KO or OE:

TMEM135 KO looks like PEX16 KO in terms of the mitochondrial fusion phenotype. Are peroxisomes OK in these KO cells?

Is TMEM135 a PPARgamma target gene?

Link to peroxisomal lipids

Fig 1 e,f,g,h – perhaps these data relating to lipids could be included in the figure title?

The data showing GNPAT KD affects TMEM135 localisation is a very strong link between peroxisome lipid synthesis and TMEM135 localisation. The Authors should consider including these in the main figures.

RE: “TMEM135 Links Peroxisomes to the Regulation Brown Fat Mitochondrial Fission and Energy Homeostasis” by Hu et al.

Manuscript No.: NCOMMS-22-43318A

We thank the Editors and Reviewers for their insightful comments and feedback, which have enabled us to enhance the quality and clarity of our manuscript. We have carefully reviewed and addressed all the concerns and suggestions raised by the three reviewers, and have incorporated new experiments and revised text accordingly. Major changes include the following:

1. To improve the quality of our imaging data, we have repeated most of the microscopy experiments and captured images using the Zeiss LSM 880 Airyscan Two-Photon Confocal Microscope. Furthermore, we have added scale bars to all micrographs to improve their interpretation. New images include, **Figures 1b, 1c, 1h, 3e, 5b, 6a, 6f, 7j, Extended Data Figures 2c, and extended data 7a.**
2. To assess subcellular localization of TMEM135 and improve the quality of results, we have repeated the subcellular fractionation and ran the whole cell lysate next to the different fractions (**Extended Data Fig. 2f**).
3. Using an orthologous approach (subcellular fractionation), we demonstrate that the mitochondrial presence of TMEM135 increases in response to NE treatment (**new Fig. 6c**), supporting of imaging data.
4. We have provided detailed methodological information, including a description of the procedures for mitochondrial isolation, mass spectrometry-based proteomics, subcellular fractionation, immunoprecipitation and GST-pull down assays.
5. To improve the accuracy of our data interpretation, we have reanalyzed our glucose tolerance tests (GTTs) and insulin tolerance tests (ITTs) using the area under the curve method.
6. To improve the clarity of our Western blot data, we have added migration information for molecular weight standards to every blot.

As a result of these changes, we believe that this is a much improved manuscript. Our point-by-point response to the reviewers' comments are provided below.

Point-by-point responses:

Reviewer's Comments:

Reviewer #1 (Remarks to the Author)

Hu and colleagues explores the impact of deficient peroxisome biogenesis (via the deletion the Pex16 gene) on mitochondria morphology. Through a series of experiments, the authors demonstrate that Pex16 deletion leads to a highly fused mitochondrial network and postulate that this is due to altered synthesis of glycerophospholipids. This prevents the translocation of TMEM135 from the peroxisomes to the mitochondria, which in turn is necessary to engage the mitochondrial fission process.

The topic and concepts presented are really interesting. Through the manuscript, the authors use an extensive combination of primary cultures, genetically engineered mouse models and molecular biology analyses. The manuscript is well written and easy to read. Nevertheless, there are some

aspects that deserve attention. The work often spreads too thin and some aspects remain a bit preliminary. Also, the manuscripts lacks consistency in some of its approaches and the description of the experiments and methods could be more precise. Some suggestions to improve the manuscript can be found below:

Response: We thank the reviewer for the insightful and encouraging comments on our manuscript. In accordance with the reviewer's suggestions, we revised the manuscript as detailed below.

1/ In Figure 1C, the authors argue that Pex16 deficient adipocytes show compromised fission. This, however, is difficult to evaluate. Even if fission occurred, the dye will not dilute unless there are other fusion events. Rather fact that the green fluorescences also argues for impaired fusion. The difficulty to evaluate this aspect also comes from the fact that changes in the amount/intensity of fluorescence are, at least to me, not possible to judge from the images. Some quantifications could help.

Response: Thank you for your suggestion. We repeated the experiment to obtain higher resolution images in order to evaluate mitochondrial fission and fusion events and quantify changes in the amount/intensity of fluorescence. These quantifications are included in the **revised Figure 1c**. We agree with the reviewer that it is likely that mitochondrial fusion is also blocked in Pex16 KO adipocytes in addition to the impaired fission and have revised the manuscript with the following discussion on page 5:

“As mitochondria fuse, the GFP signal diffuses throughout the mitochondrial network, leading to dilution of the fluorescence signal. This occurs in control cells, indicating that mitochondrial fusion is occurring normally in these cells. In contrast, the knockout cells show a small area of green fluorescence and minimal dilution of fluorescence (**Fig. 1c**). The presence of green fluorescence suggest that PAGFP is present in the mitochondria, which would normally result in yellow fluorescence if the TMRE-labelled mitochondria fused with other mitochondria. The absence of yellow fluorescence and the lack of fluorescence dilution suggest that mitochondrial fusion is impaired in these cells, presumably because mitochondria are already elongated due to impaired fission. Quantitative analysis of the images indicated that Pex16 KO significantly decreased mitochondrial fusion (**Fig. 1c**). Thus, the appearance of tubular mitochondria in Pex16 KO adipocytes is likely not due to increased fusion, but rather impaired fission of mitochondria.”

2/ The authors repeatedly observe very drastic changes in mtDNA content upon alterations of Pex16, GNPAT or TMEM135 levels. This aspect is extremely important for the phenotypes observed, yet a clear explanation is absent. The authors speculate that this could be related to deficient fission, yet never really demonstrate this point. It would be important to, at least, demonstrate that there is impaired mtDNA replication and explore potential alterations of players involved in DNA replication processes, such as TFAM. The authors might also want to discuss this finding in the perspective of other models of impaired fission or enhanced fusion. Finally, it would be beneficial to confirm such drastic changes by using two additional targets for their mtDNA vs. nDNA measurements.

Response:

Thank you for these suggestions. Consistent with the decreased mtDNA content, **new Extended Data Figure 4b** shows that TMEM135 KO brown adipocytes have significantly decreased gene expression of the mtDNA-encoded genes MT-CO1 and MTND6, while nuclear-encoded genes for mitochondrial proteins (e.g. UCP1) and factors involved in mitochondrial biogenesis (Tfam and Nrf1) were unchanged.

As suggested by the reviewer, we have added the following discussion on implications of these results on page 5 of the revised manuscript:

“The distribution of mtDNA within cells depends on continuous division and fusion events that are responsive to the specific needs of the cell type²⁴. Maintaining a balance between division and fusion is important for proper distribution and maintenance of mtDNA. Inhibition of mitochondrial fission results in mitochondrial elongation, along with defects in oxidative phosphorylation and mtDNA loss during cell division.”

3/ Some measurements were done, out of the blue, on cold-exposed mice rather than in mice under normal housing conditions (e.g.: Extended Data Fig.1, proteomic analyses in Fig.2, Fig.4F). The authors do not provide a clear rationale for this choice and the results might highly be biased by the treatment. Could the authors do some test to corroborate whether the observation is also occurring in mice at normal housing temperature?

Response: Our previous studies indicate that BAT mitochondria are more elongated at room temperature and undergo fission in response to cold exposure in mice (Park et al., JCI 2019). However, mitochondria in cultured brown adipocytes tend to be constitutively fragmented, perhaps because cell culture represents a nutrient surplus condition under which mitochondria are generally more fragmented and exhibit reduced bioenergetic efficiency. Cold-induced mitochondrial fission in BAT is blocked in Pex16-AKO mice. The mice did not have a phenotype at room temperature. Thus, to understand the mechanism through which peroxisomes regulate mitochondrial dynamics, we characterized changes in the BAT mitochondrial proteome resulting from peroxisome deficiency in cold-treated mice. The rationale for cold treatment is now discussed on page 6 of the revised manuscript.

4/ For the proteomic analyses on mitochondrial fractions, the authors should provide some analyses to certify the purity (or clear enrichment) of the mitochondrial fraction.

Response: To assess the purity/enrichment of the mitochondrial fraction, we performed western blotting in whole cell lysate and mitochondrial fraction of mice using antibodies specific to mitochondrial marker (COX4), nucleus marker (LaminA/C), ER marker (PDI), peroxisome marker (PMP70) and lysosome marker (LAMP2). These results are presented in the **new Extended Data Fig 2a**.

5/ The authors argue that Pex14 and Phyh have mitochondrial presence in Pex16 KO mice. However, this is not proven. It could be either marginal/immature peroxisomal content or contamination of other fractions. Can the authors support this statement with some imaging data?

Response: To validate our proteomics results suggesting that peroxisome deficiency leads to mislocalization of certain peroxisomal proteins to mitochondria, we stained Pex16 KO and control brown adipocytes expressing mito-GFP with an antibody against Pex14. As shown in **new Extended Data Fig 2c**, Pex16 KO resulted in increased localization of Pex14 in mitochondria.

6/ The authors should evaluate the localization of TMEM135 through cellular fractionation techniques in order to better estimate the fraction of the total TMEM135 content that resides in the mitochondria.

Response: We appreciate this suggestion. **Revised Extended Data Fig. 2f** shows Western blot analysis of different subcellular fractions run alongside whole cell lysates to better estimate the fraction of the total TMEM135 content present in mitochondria.

7/ In most imaging experiments the TMEM135 signal is difficult to interpret and the experiments would definitely benefit from some quantifications. A second aspect that could be improved is that large areas of these cells are lipid droplets. Maybe not in all experiments, but at least in the most critical ones, visualization of lipid droplets could provide a more accurate view of the cellular organelle distribution.

Response: TMEM135 exhibits punctate staining associated with mitochondrial markers presumably because it is localized only at the site of mitochondrial fission where Drp1 is recruited, thus preventing indiscriminate mitochondrial fragmentation. To validate the TMEM135 immunofluorescence signal, we stained BAT from cold-treated control and TMEM135-AKO mice using the TMEM135 antibody, which was purchased from Aviva Systems Biology Corporation. As shown in **Figure A** below, TMEM135 KO resulted in loss of the antibody signal, confirming the specificity of this antibody. As per your suggestion, the revised manuscript includes quantification of signal in imaging experiments.

Figure A. Immunofluorescence analysis using antibodies against PMP70 and TMEM135 in BAT of control and TMEM135-AKO after 7 days cold challenge. Scale bar: 10 μm .

We appreciate the reviewer's suggestion that visualization of lipid droplets would provide a more comprehensive view of organelle distribution in our experiments. In response to this suggestion, we knocked down TMEM135 in brown adipocytes and stained the cells with LipidTox and TMRE to examine the effects on lipid droplets and mitochondria, respectively, using confocal microscopy. These results are shown in **Figure B** below. TMEM135 KD resulted in more elongated mitochondria, consistent with the results reported in the manuscript. Moreover, the KD appeared to result in mitochondria being more closely associated with lipid droplets. Previous studies (PMID: 30905670) indicate that peridroplet mitochondria have unique properties with respect to bioenergetics and dynamics. We are intrigued by the potential impact of TMEM135 inactivation on lipid droplet/mitochondria interaction and plan to investigate this further in future.

Figure B. Mitochondrial morphology analysis using Zeiss LSM 880 Airyscan Confocal Microscope brown adipocytes stained with TMRE and LipidTOX after knockdown or overexpression TMEM135. Scale bar: 5 μm .

8/ Does FCCP induce mitochondrial fission in TMEM135 KD cells?

Response: In order to address this question, we treated TMEM135 KD brown adipocytes with 1.5 μM FCCP for 1h. As shown in **Figure C** below, the chemical uncoupler did not affect mitochondrial fission in TMEM135 KD cells.

Figure C. Mitochondrial morphology analysis using Zeiss LSM 880 Airyscan Confocal Microscope in brown adipocytes treated for 1h with 1.5 μM FCCP or vehicle after knockdown of TMEM135. Scale bar: 5 μm .

9/ In Figure 3H, the AA+R value should be subtracted all along the exp. Based on the results presented, the differences in respiration might be blunted once corrected by AA+R values. Second, the decrease in respiration in TMEM135 KD cells upon FCCP treatment suggests that mitochondria might be more sensitive to it. Could the experiment be repeated with a lower FCCP concentration?

Response: Thank you for your suggestion. In the **revised Fig 3h and Fig. 7e**, we subtracted the AA+R value. In the original manuscript, we inadvertently wrote wrong concentrations of Seahorse assay reagents, including the FCCP concentration, which was off by an order of magnitude. On pages 29 of the revised manuscript, the correct concentrations are provided as following:

“For the assay, cells were treated sequentially with final concentration 1 μM oligomycin, 1.5 μM FCCP (carbonyl cyanide-p-trifluoromethoxyphenylhydrazone), and 1 μM rotenone plus 1 μM antimycin A.”

10/ Figure 4G there is a clear sex dimorphism in the influence of cold exposure on core body temperature. Is this dimorphism also observed in mitochondrial architecture or mtDNA content?

Response: We appreciate the reviewers' suggestion regarding the potential influence of sex on mitochondria. In TMEM135-AKO mice, both male and female animals have decreased core body temperature (**Fig. 4g** and **Extended Data Fig. 4d**), but the magnitude of the effect differs between the two sexes. To determine if dimorphism exists on mitochondrial architecture, we isolated stromal vascular fractions from a female TMEM135-AKO mouse brown adipose tissue and immortalized them. We then differentiated the immortalized cells into adipocytes and subjected them to imaging using the Zeiss LSM 880 Airyscan Confocal Microscope. Our results indicate that, genetic inactivation of TMEM135 in female cell (**Fig. D** below) results in elongated mitochondrial morphology, similar to the results obtained using males cells (**Fig. 3e**).

Figure D. Mitochondrial morphology analysis using Zeiss LSM 880 Airyscan Confocal Microscope in brown adipocytes isolated from female control and TMEM135-AKO mice. Scale bar: 5 μ m

Although there appears to be no difference in mitochondrial morphology in male vs female cultured adipocytes, our metabolic phenotyping revealed that TMEM135 deficiency is associated with some sex differences (e.g. BW), while other phenotypes, such as GTT/ITT are similar. We believe that sexual dimorphism is common in mice and is not unique to the TMEM135 model.

11/ At least for me, it is difficult to make any conclusion from Fig.6A. The mitochondrial architecture in these images is impossible to visualize. Peroxisomes seem as big as mitochondria, and some look like actually exactly the same detection element. This problem extends all along figure 6 and Extended Data Figure 6, where all imaging studies should be quantified.

Response: To address this comment, we repeated the experiment in Fig. 6a by staining BAT with 2 separate Tomm20 antibodies: Proteintech (# CL594-11802) and CST (# 42406S). Both antibodies exhibited punctate staining, as reported by multiple others groups using BAT or brown adipocyte samples (see for example, PMID: 24713652 and PMID: 33956354). Thus, while Tomm20 staining might reveal mitochondrial architecture in certain cell types, in brown fat it tends to result in punctate staining. Importantly, our new staining more clearly demonstrates association of peroxisomes and mitochondria in BAT of cold-treated mice. These results are presented in **revised Fig. 6a**.

Additionally, we performed quantifications of Figure 6 and Extended Data Figure 6 to provide a more detailed and accurate description of the changes in imaging. Please see quantifications of **Figure 6a, b, f, Extended Data Figure 6a** and **new Extended Figure 7a**.

12/ Given the relation of TMEM135 on thermogenesis, could the authors evaluate if its knock-down or overexpression alters adrenergic signaling?

Response: We appreciate the reviewer's comments regarding the potential influence of TMEM135 on adrenergic signaling and its implications for thermogenesis. This is an area of research we are

interested in pursuing for future research, but believe that it is beyond the scope of the current study. Thank you for your helpful suggestion, which has helped us to expand the scope of our research and identify promising areas for future investigation.

Minor:

1/ Figure 1a – the experimental design should be clarified. Are these adipocytes from Pex16 floxed mice that are infected with the Cre recombinase or are they from Pex16 floxed mice (control) and Pex16 AKO mice?

Response: These adipocytes were isolated from Pex16 floxed mice (control) and Pex16 AKO mice (Pex16 KO). The figure legend has been updated to clarify this point.

2/ Figure 1e - the enzymes names should be spelled out in the legend

Response: We have spelled out the enzyme names in Figure 1e legend.

3/ Figure 1f/g - data for GNPAT protein levels should be provided

Response: Western blot analysis of protein levels in GNPAT KD brown adipocytes is shown in **new Fig. 1g**.

4/ Extended Data Figure 2D: the images presented are not really convincing. First total homogenates should be run next to the different fractions in order to appreciate enrichments. Second, some bands look really weird and don't really align. Finally, some of the outcomes are unclear (Akt presence in the mitochondria). I would strongly encourage the authors to repeat the experiment taking into account the points above.

Response: Thank you for this suggestion. We repeated the subcellular fractionation and ran the whole cell lysate (WCL) next to the different fractions according to your suggestion. The new blot presented in **Extended Data Fig. 2f** has a much better quality. Furthermore, the contaminating Akt band is no longer seen in the mitochondrial fraction, presumably because in the initial experiment, the mitochondrial pellet was not washed well.

5/ Extended Data Figure 4b: why is there such little uncoupling in these mitochondria? Could this be linked to the fact that Complex II, instead of Complex I is being tested?

Response: Thank you for requesting this clarification. Respiration linked to Complex II, instead of Complex I, was measured. This experiment involves the use of frozen brown adipose tissue isolated from control and TMEM135-AKO mice following a recently published novel approach to measure mitochondrial respiration in frozen biological samples (PMID: 32432379). Succinate was used as a

substrate of Complex II, while rotenone was used to inhibit Complex I. These are now presented in **Extended Data Fig. 4c**.

6/ Extended Data Figure 4m: could the authors please provide absolute blood glucose values?

Response: This should be Figure 4m since Extended Data Figure 4 does not have panel m. We reanalyzed our data by using absolute blood glucose values according to your suggestion. In addition, we measured the area under the curve (AUC). These results are presented in **revised Fig. 4m**.

7/ Figure 5E: This result is surprising, as thermoneutrality generally takes longer (around 4 weeks) to have such a strong impact on BAT-related thermogenesis. Could the authors please discuss how come the wild-type mice are so impaired and clarify whether these mice were males or females?

Response: In our study, we maintained male WT and TMEM135^{TG} at thermoneutrality (30°C) for 12 days prior to subjecting them to acute cold exposure. This information has been corrected in the revised manuscript. The 12-day warm room treatment was sufficient to induce severe cold intolerance in WT mice. This is consistent with previous studies suggesting that 10 days of thermoneutrality results in a marked decrease in UCP1 in brown and beige fat in mice (PMID: 27568548).

8/ Figure 5F: when compared to the TMEM135 KO mice, it seems that it is the controls who actually change, being much heavier in this intervention. Can the authors at least discuss that?

Response: Thank you for requesting this clarification. As noted in the figure legend of the revised manuscript, the results in **Figure 5F** are from male mice, which are more susceptible to HFD-induced obesity. In contrast, the TMEM135 KO body weight data in **Figure 4J** are from female mice.

9/ Extended Data Fig.5E: O2 and VCO2 values should be corrected by lean mass.

Response: Thank you for your suggestion. We have corrected **Extended Data Fig.5e and f** according to your suggestions.

10/ Extended Data Fig.5H: This is a very preliminary experiment, which departs quite significantly from the techniques used in this or other papers to evaluate metabolic flexibility. Proving this point would take many more measurements, and therefore, I would instead suggest to remove these two panels.

Response: We appreciate the reviewer's concern with our methodology for measuring ambient CO₂ in rodent home-cages, but feel that our experiment is justified to include. Regarding the claim that this experiment was preliminary, we tested 14 wild-type and 13 transgenic animals, in an experimental design that was powered to detect a 10% change in ambient CO₂ (based on power calculation done here: <https://www.stat.ubc.ca/~rollin/stats/ssize/n2.html>). We chose this power based on our vCO₂ measurements using traditional indirect calorimetry chambers presented in Extended Data Figure 5e-f,

where we detected a non-significant ~15-20% change in vCO₂ between the groups. We therefore do not think our data is preliminary from the standpoint of statistical power.

We also understand the reviewer's concern over this new approach to measuring differences in CO₂ production, but believe our controls are adequate here. First, in this same figure we presented data from these same lines of mice in traditional indirect calorimetry chambers, noting similar trends in the data. Head-to-head comparisons like this are critical for evaluating new methods, as we have done here. Second, we compared the transgenic animals to a control group of wild-type mice, noting an expected reduction in CO₂ production in wild-type animals as they adapt to thermoneutrality, but not seeing this adaptation in the transgenic animals.

Finally, we'd like to explain why we developed this approach for measuring CO₂, and what benefits it may have for other researchers. Measuring ambient CO₂ allowed us to make measurements at thermoneutrality, something our indirect calorimetry system is not equipped to do. This revealed the lack of adaptation to thermoneutrality in the transgenic animals, a phenomenon that would not have been observable at room temperature. In addition, we are able to measure CO₂ in their normal home-cages, without the added stress of moving mice into indirect calorimetry chambers. This may be useful to other researchers who wish to avoid the stress of switching chambers, or would like to perform longer measurements over weeks or months, which are not suitable to indirect calorimetry chambers. For these reasons, we believe it is appropriate to keep these experiments in the paper. Previous studies from Dr. Alexxai Kravtiz, one of the co-authors, reported the use of these sensors to measure activity (PMID: 36577735).

11/ Figure 5M: why is the Akt band lower iinsulin-treated mice? In general, insulin treatment should lead to a slightly higher band.

Response: It is unclear why the total Akt band is slightly lower in insulin-treated mice. The exact same membrane from the pAkt blot was stripped and reprobed with the total Akt antibody. The Akt band in the TG samples is slightly higher, possibly reflecting some post-translational modification of the protein associated with TMEM135 overexpression. Alternatively, the difference in size is likely due to an artifact.

12/ Extended Data Figure 5I: is this insulin stimulated or basal state? Also, shouldn't the results in BAT be corrected by the fact that it has a lower lipid content per gram of tissue?

Response: The PET/CT data are from mice in the basal state. The results are normalized per gram of tissue for BAT as is for other tissues.

13/ How would these markers look in the basal state?

Response: We presented in the Extended Data Figure 5I is basal state.

Reviewer #2 (Remarks to the Author)

Mitochondria are dynamic organelles that undergo fission and fusion. Fission and fusion of mitochondria control their metabolic activity, including with regards to their thermogenic capacity in brown adipocytes. The metabolic activity of peroxisomes also influences the thermogenic activity of brown adipocytes. Here, Hu and coauthors present evidence that the peroxisomal membrane protein TMEM135 is a regulator of mitochondrial fission and thermogenesis. Thermogenic stimuli promote association between peroxisomes and mitochondria and plasmalogen (specialized lipids made in part by peroxisomes)-dependent localization of TMEM135 to mitochondria. Knock out of TMEM135 in brown adipose tissue in mice blocks mitochondrial fission, impairs thermogenesis and increases diet-induced obesity and insulin resistance, while overexpression of TMEM135 counteracts these effects. At mitochondria, TMEM135 is involved in the retention of the fission factor, Drp1.

This manuscript describes an interesting communication pathway between peroxisomes and mitochondria mediated by the movement of the membrane protein TMEM135 from the peroxisome to the mitochondrion in brown adipocytes to regulate the thermogenic activity of brown adipose tissue. The findings presented in this manuscript will represent an important contribution to our understanding of the control of the metabolic activity of brown adipose tissue and the role played in this process by communication between peroxisomes and mitochondria once the authors have addressed a number of critical issues as outlined below:

Response: Thank you for your positive feedback and insightful comments on our manuscript. We appreciate your recognition of the significance of our findings on the role of TMEM135 in regulating mitochondrial fission and thermogenesis in brown adipose tissue, and for your suggestions for further improvement. We have carefully considered your comments and have made several revisions to address the critical issues you raised. In accordance with your suggestions, we revised the manuscript as detailed below.

Principal Issues:

1) The fluorescence microscope images purporting to show fusion or fission of mitochondria have to be dramatically improved. While an image such as Figure 1b clearly demonstrates fusion of mitochondria in Pex16 KO cells versus control cells, most fluorescence images do not convincingly show either fusion or fission of mitochondria as the authors conclude one way or the other throughout the manuscript. Since these microscope images form the foundation of most of the conclusions of this manuscript, they must be dramatically improved to convincingly support the conclusions of the authors.

Response: Thank you for your suggestion. We repeated this experiment and performed the image analysis using a high resolution Zeiss LSM 880 Airyscan confocal microscope. The improved images and quantification are presented in **revised Figure 1b**.

2) Scale bars must be added to all micrographs. Given that images presented in figures are of many different sizes, simply stating that original images were taken, for example, at 60× magnification is not particularly helpful.

Response: We added scale bars to all imaging experiments in revised manuscript.

3) The authors should present fluorescence microscopy images of mitochondria corresponding to the electron microscopy images of Figure 4d and Extended Data Fig. 6e.

Response: We used mito-GFP or TMRE staining to assess mitochondrial morphology in cultured brown adipocytes. For visualization of mitochondria shape in mouse BAT, we used electron microscopy. Unfortunately, immunofluorescence analysis using various antibodies against mitochondrial markers did not provide shape of mitochondria in BAT.

4) Figure 1c. Why is there green mitochondrially targeted photoactivatable GFP (mtPA-GFP) seen in the PEX16KO cells? If indeed the mtPA-GFP is targeted to mitochondria it should present exclusively as yellow due to colocalization with the mitochondrial marker, TMRE. Please explain.

Response: Thank you for requesting this clarification. A small region of photo-activated mitochondrial network could appear green due to lack of mitochondrial fusion and photo-bleaching of TMRE, which is light sensitive. As discussed in response to reviewer 1 comment #1, the presence of green fluorescence suggests that PAGFP is present in the mitochondria, which would normally result in yellow fluorescence if the TMRE-labelled mitochondria fused with other mitochondria. The absence of yellow fluorescence and the lack of fluorescence dilution suggest that mitochondrial fusion is impaired in Pex16 KO cells, presumably because mitochondria are already elongated due to impaired fission.

5) Why is there a TMEM135 band (of the same size as found in the Control) in the western blot of the brown adipose tissue coming from the TMEM135-AKO mouse?

Response: The residual TMEM135 band in the TMEM135-AKO samples likely reflects incomplete Cre-mediated recombination since adipose tissue is a mixture of adipocytes and other cell types.

6) A figure showing the structure of the entire mouse TMEM135 gene would help guide the reader.

Response: The structure of the mouse TMEM135 gene is provided in the revised Figure 4a.

7) The migrations of molecular weight standards must be added to every western blot.

Response: We have added migrations of molecular weight standards to every western blot in the revised manuscript.

8) Extended Data Fig. 6a. Pex16-KD cells, not Pex16-KO cells.

Response: Differentiated SVF cells from Pex16 floxed mice (control) and Pex16 AKO mice (Pex16 KO) were used in this experiment. The figure legend has been corrected to clarify this point.

9) *Figures 6e and 6f. TMEM135OE not simply TMEM135.*

Response: Thank you for this suggestion. We have corrected that to TMEM135 OE in **Figures 6e and 6f** in revised manuscript.

10) *Figure 6f showing recruitment of Drp1 to mitochondria by overexpression of TMEM135 is unconvincing. New images are needed, along with quantification.*

Response: New images and quantification are now provided in **revised Fig. 6f**.

Minor Issues:

1) *p. 7, line 174. ‘...five predicted transmembrane domains...’.*

Response: Thank you for this correction. This has been corrected on page 7 of the revised manuscript

2) *p. 8, line 197. ‘Extended Data Fig. 3a’ NOT ‘Extended Data Fig. 3e’.*

Response: We have corrected that to “Extended Data Fig. 3a” in the revised manuscript.

3) *p. 8, line 197. For the non-expert, what does the addition of norepinephrine do to brown adipocytes?*

Response: Thank you for this suggestion. We have revised the text on page 8 to include the following brief explanation of the role of norepinephrine:

“...norepinephrine (NE), a β -adrenergic receptor agonist that promotes thermogenesis by activating lipolysis and increasing thermogenic gene expression, further increased the expression of TMEM135...”

Reviewer #3 (Remarks to the Author):

Hu et al describe a role for the protein TMEM135 in the regulation of brown fat mitochondrial fission and, as a result, thermogenesis in mice. They propose that this protein provides a link between peroxisomes and mitochondrial – peroxisomes are known to regulate mitochondria morphology via specific lipid species. Hu et al. propose that TMEM135 forms part of this pathway. The Authors provide strong evidence that loss of TMEM135 in adipose tissue in mouse increased susceptibility to obesity and glucose intolerance on a HFD and decreased cold tolerance. The authors propose a model whereby TMEM135 acts a link between peroxisomes and mitochondria by physically

translocating to mitochondria to help regulate mitochondrial fission via DRP1.

The manuscript is generally clearly written, but is quite wordy and long in some sections. Also, the Authors use phrases like “remarkably” quite frequently, which in most cases is not needed (obviously this is a personal preference).

The data showing a key role for TMEM135 in brown fat function and energy metabolism is convincing, in general. I have more reservations over the molecular details provided in the manuscript, largely due to 1) a real lack of methodological details, 2) a lack of orthologous approaches to show TMEM135 translocation to mitochondria, for example, 3) a lack of unbiased image analysis and 4) representative images that are generally highly overexposed and also do not obviously match the authors interpretation.

I have split my review into two sections – in vivo data and cell/molecular data. Note that my expertise falls more on the side of the cell/molecular data.

Response: Thank you for your thorough and insightful review of our manuscript. We appreciate your feedback and suggestions for improvement. The manuscript has been edited to decrease wordiness and the use of phrases such “remarkably”. We have carefully considered your comments and addressed them below in our point-by-point responses.

In vivo

The increase in TMTM135 with cold is stark. Does this match other peroxisome proteins, or is this unique to TMEM135? In addition, the protein increase looks much more than the mRNA – is this correct? Is this of interest?

Response: As we reported previously (PMID: 30511960), the expression of other peroxisomal proteins, such as PMP70, Pex16 and Pex19, also markedly increases in BAT of cold-treated mice. With regard to the large increase in TMEM135 protein levels, it is possible that the TMEM135 protein increase is greater than the mRNA because the protein continued to accumulate during the seven day cold treatment while the mRNA levels was increased more acutely and then returned closer to baseline. As discussed in response to your comment below, we plan to pursue future research on transcriptional and post-transcriptional regulation of TMEM135 expression.

Cold stress test data –the data within the paper appear to be inconstant. In Fig. 5, the WT mice decline in temperate across 6 h, reaching 30 oC after 6 hours. In Fig. 4G and EDFig. 4C, control mice plateauing at 34oC after 1-2 h cold. Are data from this assay usually this variable? Is the difference due to how the mice were housed prior to the cold test?

Response: Yes, the difference is due to how the mice were house prior to cold exposure. In **Fig. 4g** and **Extended Data Fig. 4d**, the mice were directly subjected to cold treatment. As noted in the figure legend, the mice from **Fig. 5e** were maintained at thermoneutrality (30°C) for 12 days prior to acute cold exposure.

For GTT and ITT data: GTT and ITT data are typically analyzed using an area under/over the curve to describe the entire response to glucose/insulin over the test. However, the analyses are a little inconsistent here with AUC shown for the ITT and not GTT. I would suggest that AUC should be provided for GTT (corrected for 0 min) and area over the curve for ITT.

In addition, presenting data from ITT as % of initial glucose can be misleading, as it doesn't show the difference in initial blood glucose. Please see PMID: 34117483. In general, the blood glucose at 0 min GTTs shows very little difference in blood glucose, so I would expect this % analysis to be unnecessary (assuming the same length fast). Please plot raw data for ITTs.

In Fig. 4M – The Authors interpret these data as evidence for insulin resistance. The increase in blood glucose at 15 min is likely a stress response, as an increase with insulin is not expected. This is the only time point that shows a difference, so its not very clear there is an insulin resistant phenotype in these AKO mice from this test. Perhaps an AOC analysis would provide more insight as the control mice have a generally lower glucose over the course of the experiment. Whether or not these mice are insulin resistant as measured by ITT is not critical to the paper, but this needs to be interpreted correctly.

Response: Thank you for your valuable and constructive comments. According to the paper mentioned by the reviewer, “The correct way to assess the area of a GTT or ITT curve is to measure the area under the curve (AUC) and subtract the area under the baseline.” Since the baseline glucose values were similar between genotypes for all of the GTTs and ITTs presented in the manuscript, we believe that AUC is appropriate for analysis of these results. Thus, we plotted raw data for ITTs and provided AUC for all GTTs and ITTs. Please see **revised figures 4k, 4m, 5j, and 5l and Extended Fig 4g, 4h, 5i and 5i**.

With regard to **Fig. 4m**, the data are based on 11-12 female mice/group and all mice were equally treated and acclimated to the testing environment. Our conclusion that the TMEM135-AKO mice are insulin resistant also took into consideration the large increase in fasting serum insulin levels in the knockout mice as compared to the control animals (**Fig. 4l**). **Revised Fig. 4m** presents the raw data and includes AUC.

How was the NE, insulin and glucose data calculated. Was this on BW or lean mass? Could the changes in BW be a factor in improved glucose tolerance, for example, if dosed on BW?

Response: In these experiments, the dosing was based on BW, as is a common practice in animal studies. The rationale behind this is that body weight is a more practical and easy-to-measure metric, and is a reasonable proxy for overall metabolic demand. It is noteworthy that male TMEM135-AKO have impaired glucose tolerance (**Extended Fig. 4g**) despite a lack of difference in body weight (**Extended Fig. 4f**).

How was food intake measured? Was this measured as a group, or in singly housed mice? How long was food intake measured for?

Response: For food intake measurement, we singly housed mice and allowed them to acclimate for 2 days. Daily food intake was measured by adding a known amount of food and subtracting the weight of uneaten food and any crumbs for at least three days.

Molecular/Cell:

Mitochondrial proteomics:

The details on how mitochondria were isolated before proteomics is not provided in any detail, and, as far as I can see, are not provided in the reference the Authors point us towards for a detailed method (REF 19). Please include the experimental details here. In general, the proteomics data are not described in detail, and more detail could be provided. For example, “Many proteins either increased or decreased as a result of peroxisome deficiency” – the authors should provide exact numbers here.

Do other peroxisomal proteins go down in the proteomics data? The authors suggest that some peroxisomal proteins are increased in mitochondria in peroxisome deficit cells due to “mistargeting” to mitochondria. Do the Authors have evidence that these proteins now localise to mitochondria by an orthologous method? Are these the only peroxisome proteins that went up – is TMEM135 the only peroxisome protein that went down?

Response: We appreciate your suggestions and have added the following detailing on isolation of mitochondria in Methods section under *Mitochondria Isolation* (page 20):

“BAT depots were collected from Pex16-AKO and control mice after perfusion using PBS. The tissues were cut into small pieces and homogenized in 100 μ l ice cold homogenization buffer (0.25 M sucrose, 20 mM HEPES in distilled H₂O) containing a protease and phosphatase inhibitor cocktail (Sigma, P8465 and P0001) for 10 strokes in a dounce homogenizer. The pestle was washed with 800 μ l of homogenization buffer to remove any residual homogenate. The homogenates were transferred to 1.5 ml tubes and vortexed thoroughly before centrifugation at 800 \times g for 10 min at 4°C. The upper fat cake was removed by forceps and the supernatant (~800 μ l) was transferred to a new tube and centrifuged at 10,000 \times g for 10 min at 4°C. The supernatant was discarded and the pellet (mitochondrial fraction) was washed one time and then used immediately for mass spectrometry-based proteomics or stored at -80 °C for future use.”

A detailed method for mass spectrometry is described on pages 20-21 of the revised manuscript.

To address your other comments, we added the following text on page 6 of the revised manuscript:

“Since mitochondria isolated by differential centrifugation are frequently contaminated with other organelles, especially peroxisomes²³, we assessed the purity of the mitochondrial fraction by Western blot analysis using antibodies against different organelle markers. Our results show that the mitochondrial fraction was enriched in the known mitochondrial protein COX4 and had only a trace amount of the peroxisomal marker PMP70 or the nuclear marker Lamin A/C (**Extended Data Fig. 2a**). A complete list of the proteins identified by mass spectrometry in the two groups is presented in **Supplemental Table 1**. Gene Ontology (GO) analysis confirmed enrichment of mitochondrial proteins (**Extended Data Fig. 2a**). A heatmap of the proteins increased or decreased in the mitochondrial fraction as a result of Pex16 knockout is presented in **Fig. 2b**. A total of 139 proteins were significantly

decreased and 182 proteins were significantly increased in mitochondria as a result of peroxisome deficiency. Increased proteins included certain peroxisomal proteins, such as the peroxisomal biogenesis factor Pex14 and the branched chain fatty acid oxidation enzyme Phyh, which were mistargeted to mitochondria in the absence of peroxisomes (**Fig. 2c**). The mislocalization of Pex14 to mitochondria in Pex16 KO brown adipocytes was confirmed by immunofluorescence analysis (**Extended Data Fig. 2c**). Certain peroxisomal proteins were also decreased in Pex16-AKO mitochondria, including PMP70 (ABCD3) and Acox1, which our previous studies indicate are degraded in the absence of peroxisome¹⁹. The protein most overall decreased in the knockout mitochondria was a transmembrane protein called TMEM135 (**Fig. 2c**), which was originally identified by mass spectrometry as a protein enriched in peroxisomes isolated from liver and kidney and named PMP52^{21,22}.”

Organelle preps

There is no detail on how the organelle preps in EDFig2D were performed. Please provide details. This information is critical for interpreting these data. For example, are ER proteins present in the Nuc fraction? Could it be that TMEM135 is in the ER, not the Nucleus? Also, ref 19 has a similar blot but this blot has obvious peroxisomal contamination in the mitochondrial fraction. Here, the preps look very clean, despite it being very challenging to separate mitochondria and peroxisome by differential centrifugation. Nevertheless, it does look like TMEM135 is more multilocalised between mitochondria and peroxisomes than PMP70 – what about other peroxisome proteins? In addition, contrast and banding on this blot are strange – for example AKT bands do not align with the label. Would an organelle IP approach provide a purer method to separate mitochondria and peroxisome proteins?

Response: Thank you for your insightful and constructive comments. We now provide the following methodological details for organelle isolation on page 21 of the revised manuscript:

Subcellular Fractionation of Brown Adipocytes

“Peroxisomes were isolated from cultured brown adipocytes using Peroxisome Isolation Kit (Sigma-Aldrich) with modifications of the manufacturer’s protocol. All reagents were prechilled on ice prior to the start of the assay and all steps were performed on ice to preserve organelle integrity. Briefly, 4×10^7 brown adipocytes were washed with PBS and resuspended in 1X Peroxisome Extraction Buffer (5 mM MOPS, pH 7.65; 0.25 mM sucrose, 1 mM EDTA, and 0.1% ethanol). The cells were broken using a dounce homogenizer and centrifuged at $1,000 \times g$ for 10 minutes to collect the nuclear pellet. The supernatant was then transferred to a new tube and centrifuged at $10,000 \times g$ for 10 minutes to collect the mitochondria-enriched pellet. The supernatant was transferred to a new tube and centrifuged at $21,000 \times g$ for 40 minutes to collect the crude peroxisome fraction (CPF). The supernatant was transferred to a new tube as the cytosol fraction. To isolate the peroxisome fraction, CPF was further fractionated using OptiPrep™ density gradient medium and centrifugation at $100,000 \times g$ for 1.5 hours, as described in the manufacturer’s protocol. The nucleus, cytosol, mitochondria, and peroxisome fractions were subjected to SDS-PAGE and immunoblotted using organelle-specific antibodies.”

We repeated the subcellular fractionation experiment and included whole cell lysate. Please see the revised **Extended Data Fig. 2d**, which supports our original result indicating that TMEM135 is present in peroxisomes, mitochondria and the nucleus.

With respect to potential ER vs nuclear localization of TMEM135, it is unlikely that the nuclear pellet was contaminated with ER, since the pellet was washed prior by SDS-PAGE and isolation of ER requires centrifugation at $152,000 \times g$ for 70 minutes (PMID: 26331984). Nevertheless, it is possible that TMEM135 is also partially localized in the ER (see **Fig. E** in response to your comment below). Nuclear localization is further supported by the immunofluorescence data in **Extended Fig. 2f**. The potential nuclear localization of TMEM135 is not a key aspect of the current manuscript. We are planning a future study to further investigate TMEM135 nuclear localization and its physiological importance.

With regard to the organelle IP approach, we had originally planned to use Mito-Tag to isolate mitochondria from cold-treated Pex16-AKO mice. To this end, we crossed Mito-Tag mice with our Pex16-AKO mice. However, due to poor expression of the Mito-tag in BAT, we decided to use the differential centrifugation approach instead.

Imaging data- image quality, quantification and representative images:

In general, the standard of the microscopy images makes them hard to interpret. There are often over-exposed to that the mitochondrial signal is very strong and it's hard to see individual structures. The Authors use the term "merged" for colocalisation between green and red signals, I think. This is not the correct terminology. Also, separate images should be shown. This is critical. Also, the interpretation provided by the Authors in some cases is not clearly matched by the representative images. For example – Figure 7J. I could not pick which of the images corresponded with the quantitation from viewing these representative images. This would be aided by 1) quantifying all images in an unbiased way and/or 2) blinding the analysis of mitochondrial fragmentation if this isn't how this was done already.

Response: Thank you for pointing this out. We repeated most of image experiments using a Zeiss LSM 880 Airyscan Confocal to improve quality of images and avoiding overexposure in images. We also provided separate channel for all images. In response to the reviewer comments, we asked another investigator in our lab who was blinded to the identity of the samples to quantify images in an unbiased way and our results were similar. Please see revised **Fig. 3e, 5b and 7j** for high resolution images and quantification. The term "merge" refers to the composite image of different channels and is commonly used in the literature. For further clarity, we now list the individual channels in the composite images.

Fig. 2F – The Mander's overlap coefficient. Why was this method chosen to determine colocalisation? Where analysis like this has taken place, how many cells do the 3 ns in the graph represent? This information should be provided in all cases?

Response: The Mander's overlap coefficient is a commonly used method to quantify the degree of colocalization between two fluorophores in microscopy images. This method was chosen for

determining colocalization because it is widely used, easy to unbiased interpret, and provides a robust measure of colocalization that can be used in a variety of experimental contexts. We used three independent experiments instead of cells number to do analysis and qualification.

Fig 1C – why is there a green section – this could coloclaise with the TMRE stain? Can C be quantified?

Response: A small region of photo-activated mitochondrial network could appear green due to lack of mitochondrial fusion and photo-bleaching of TMRE, which is light sensitive. The presence of green fluorescence suggests that PAGFP is present in the mitochondria, which would normally result in yellow fluorescence if the TMRE-labelled mitochondria fused with other mitochondria. The absence of yellow fluorescence and the lack of fluorescence dilution suggest that mitochondrial fusion is impaired in Pex16 KO cells, presumably because mitochondria are already elongated due to impaired fission. Quantification is now provided in the **revised Fig. 1C**.

1GH – the image looks very blurry. Can the authors provide a better one?

Response: We repeated this experiment by using the Zeiss LSM 880 Airyscan Confocal to improve quality of images. The new images are provided in the **revised Fig. 1h**.

Figure 3E - Were investigators blinded for the analysis?

Response: We were not blinded for the analysis in the original **Fig. 3e**, but have since asked a lab member who was blinded to the identity of samples to analyze images in an unbiased way and our results were similar. In the revised images for **Fig. 3e** taken using a high resolution confocal microscope, the effect of TMEM135 KD on mitochondrial morphology is very clear.

Figure 4D – the EM images show increased density around mitochondria – is this a consistent feature? What does this mean?

Response: Although the effect of TMEM135 KO on mitochondrial morphology was consistent, the staining was somewhat mosaic– there were areas that appear more electron dense (darker) and areas that appear less electron dense (lighter) within the same sample. In **Fig. 4d**, the right image was likely acquired from the area with more heavy metal staining, which is why cytosol looks electron dense and appears darker. In general, brown adipose tissue will have differences in staining pattern of heavy metals because of (1) structure/density - lipid rich samples are dense and heavy metals do not successfully penetrate in all areas of the sample during sample preparation and (2) fixation quality – periphery/ surface of the sample might be fixed better than the central part of the tissue since tissue is dense. Overall, this results in different electron density appearance in TEM – areas that have more staining with heavy metals will appear more electron dense (darker) and areas with less heavy metal staining will appear less electron density (lighter).

Figure 4F – 8 cells per condition seems low. How many different mice was this from?

Response: We used two mice for each genotype and took at least 20 images at different magnifications for each group. Elongated mitochondrial morphology of brown adipose tissue from TMEM135-AKO in TEM was consistent in different images and we picked 8 cells per condition (with each cell containing ~60-80 mitochondria) for the quantification. Quantification in Fig. 4f is based on images taken at 1000-2000x magnification. This method and the results obtained are consistent with our previously published results in Pex16-AKO mice (PMID: 30511960).

Figure 6a – TOMM20 is a mitochondrial marker, so why does this look so different from the TMRE staining in other images throughout the paper?

Response:

We repeated the experiment in **Fig. 6a** by staining BAT with 2 separate Tomm20 antibodies. Both antibodies exhibited punctate staining, as reported by multiple other groups using BAT or brown adipocyte samples (see for example, PMID: **24713652** and PMID: **33956354**). Thus, while Tomm20 staining might reveal mitochondrial architecture in certain cell types, in brown fat it can result in punctate staining. Importantly, our new staining more clearly demonstrates association of peroxisomes and mitochondria in BAT of cold-treated mice. These results are presented in **revised Fig. 6a**.

Additionally, we performed quantifications of Figure 6 and Extended Data Figure 6 to provide a more detailed and accurate description of the changes in imaging. Please see quantifications of **Figure 6a, b, f, Extended Data Figure 6a** and **new Extended Figure 7a**.

Mitophagy data – MitoKeima – ED Fig. 1B. Can this be quantified?

Response: We quantified the results for this **Extended Data Fig. 1b** according to a previous report (PMID: 26549682).

Fig 1G – this needs to be quantified in some way – there are ways to quantify the Mito network.

Response: According to the reviewer suggestion, we quantified the mitochondrial morphology. The results are presented in **revised Fig.1 h**.

Imaging data – mitochondria-peroxisomes contacts:

The data showing apparent increase in peroxisome-mitochondrial contacts with NE are not overly convincing. 6b – the GFP-PTS-1 signal doesn't seem to change much with NE, but the morphology does. Could the increase in apparent signal overlap be technical, and due to the change in what the mitochondrial signal looks like, rather than genuine contacts? – here the mito signal is very highly exposed as well, making it hard to discern if the green and red are overlapping or just in close proximity.

Overall, these imaging data need to be paired with orthogonal assays (biochemistry?) or a more

extensive imaging analysis (e.g. EM, high resolution) to really probe the interactions between peroxisomes and mitochondria.

Response: We appreciate the helpful comments. We repeated this experiment and used a Zeiss LSM 880 Airyscan Confocal, which allowed us to improve the quality of the images and avoid overexposure. Our new results (**revised Extended Data Fig. 7a**) show that mitochondrial morphology became more fragmented with NE treatment as compared to the vehicle group. Moreover, we found that peroxisome-mitochondrial contacts significantly increased, as shown in the figure and quantified in the analysis. These findings provide further support for our hypothesis that peroxisomes are involved in the regulation of mitochondrial morphology and function in brown adipocytes. We thank the reviewer for bringing this issue to our attention and helping us to improve the quality of our data.

With regard to your comment about the use of an orthologous approach, our new data using subcellular fractionation indicate that the mitochondrial presence of TMEM135 (which mediates peroxisomal regulation of mitochondrial fission) increases in response to NE treatment (**new Fig. 6c**), as described in our response to comment below.

Imaging data – TMEM135 in mitochondria:

The authors conclude, from these images in Fig. 2F, that mitochondrial localisation of TMEM135 was lower in PEX16 KO cells – but the images do not convincingly show that TMEM135 in mitochondria. If TMEM135 was in mitochondria, as the authors suggest, I would expect it to begin to look like mitochondria in morphology, not remain as signal puncta near mitochondria. It maybe that this is the case, but separate images are not shown. Data from REF 22 (elife) also not showing mitochondria localisation – but more like the association of TMEM135 vesicles/puncta close to mitochondria

In PEX16 KO cells do not have peroxisomes. So where is the TMEM135 localised? The TMEM135 is still in puncta, but this cannot be peroxisomes. Fig. 2F – there are a large number of TMEM135 puncta very distant from mitochondrial stain in the Pex16 KO image. Is this image representative? In all other images the whole cells are full of mitochondria. Why is there a large gap in this image. Is it the nucleus?

Response: We thank the reviewer for the constructive comments. In contrast to mito-GFP (or TMRE), which shows the entire mitochondrial network, TMEM135 exhibits punctate staining associated with mitochondrial markers. We believe the reason for punctate staining is that TMEM135 is likely localized only at the site of mitochondrial fission where Drp1 is recruited, thus preventing indiscriminate mitochondrial fragmentation.

In the **revised Figure 2F**, we provided separate channels for mito-GFP and TMEM135 and added DAPI in the merged image for visualization of the nucleus. In Pex16 KO cells, which lack peroxisomes, TMEM135 has less co-localization with the mitochondrial marker. DAPI staining indicated that the large gap in the image is the nucleus. To determine if TMEM135 is also localized in the ER, we stained cells with an antibody against PDI, an ER maker, and observed partial localization of TMEM135 in the ER in the absence of peroxisomes. Please see the images in **Figure E** below.

Figure E. Immunofluorescence analysis in control and Pex16-AKO brown adipocytes stained with antibodies against PDI and TMEM135. Scale bar: 10 µm.

In 6c – the BFP signal looks to be less punctate and more dispersed – it is the dispersed signal that overlaps with the mitochondria – giving the large increase in the overlap coefficient. Or is this picking up small contacts between puncta (presumably peroxisomes) and mitochondria. It would be really useful to see another peroxisome marker here – is this dispersed signal specific to TMEM135? The claimed increase in TMEM135 in mitochondria with NE shown in 6C would be more convincing if paired with orthogonal methods, such as blotting for TMEM in mitochondrial fractions.

Response: We appreciate these recommendations. Regarding the BFP signal, we agree that it appears less punctate and more dispersed in some images (now shown in **Fig. 6b**). We believe that this dispersed signal represents peroxisomes that are in close proximity to mitochondria. Following the reviewer suggestion, we isolated peroxisomal and mitochondrial fractions from brown adipocytes treated with or without NE, and performed Western blot analysis using an antibody against TMEM135. The results presented in **new Fig. 6c** shows that TMEM135 localization increases in the mitochondrial fraction after NE treatment. Together, these data are consistent with our hypothesis that TMEM135 translocates from peroxisomes to mitochondria to promote mitochondrial fission after NE treatment.

The same is true for other data in the paper, such as the in Fig 6F, which does not convincingly show more DRP in mitochondria with TMEM135 OE.

Response: Thank you for your suggestion. Please see **revised Fig. 6F** for improved images and quantification.

Interactions data (PD, IP): How were Ips and pill downs performed. What buffer was used? This is important for how we interpret these data. For example in EDFig2C – Can the Authors confirm which buffer was used for the immunoprecipitation? The methods are very unclear on this. If a sucrose buffer was used, then the interaction between TMEM135 and PEX19 is not convincing as the IP may just be pulling down whole peroxisomes. Similarly, is the flag-tag of TMEM135 cytosolic, or luminal?

Response: Sucrose was not used in buffers for interaction studies. The FLAG tag is on the C-terminal end of the protein and is expected to be cytosolic. We have added the methods for Immunoprecipitation and GST Pull-down Assays on pages 19-20 of the revised manuscript.

TMEM135 KO or OE:

TMEM135 KO looks like PEX16 KO in terms of the mitochondrial fusion phenotype. Are peroxisomes OK in these KO cells?

Response: We conducted an immunofluorescence analysis using antibodies against PMP70 and TMEM135 in brown adipose tissue of control and TMEM135-AKO mice after a 7-day cold challenge. Surprisingly, we observed an increased number of peroxisomes in TMEM135-AKO mice compared to control mice. Please see **Figure A** provided in response to Reviewer 1, comment #7. This finding is intriguing, and we plan to investigate in detail the potential role of TMEM135 in regulating peroxisomal biogenesis and function in brown adipose tissue through additional experiments.

Is TMEM135 a PPARgamma target gene?

Response: To address this question, we treated brown adipocytes with the 5 μ M of the PPAR γ agonist rosiglitazone for 24 hours and performed gene expression analysis. TMEM135 expression significantly

increased after rosiglitazone treatment, similar to AP2, a known target of PPAR γ . Thus, TMEM135 appears to be regulated by PPAR γ . We are interested in pursuing future research on transcriptional and post-transcriptional regulation of TMEM135 expression, but present the result from rosiglitazone experiment below (**Figure F**) for the reviewers.

Figure F. qPCR analysis of AP2 and TMEM135 gene expression after 5 μ M rosiglitazone treatment for 24 hours.

Link to peroxisomal lipids

Fig 1 e,f,g,h – perhaps these data relating to lipids could be included in the figure title?

The data showing GNPAT KD affects TMEM135 localisation is a very strong link between peroxisome lipid synthesis and TMEM135 localisation. The Authors should consider including these in the main figures.

Response: We appreciate your suggestions. We have revised Fig 1 title to “Brown adipocyte peroxisome deficiency results in tubular mitochondrial networks through impaired mitochondrial fission, which could be rescued by peroxisome-derived lipids.” The data from Extended Data Fig. 2e and 2f are now presented as **new Fig. 2g and 2h**.

REVIEWER COMMENTS

Reviewer #1 (Remarks to the Author):

The revised version of the manuscript shows some clear improvements compared to the original submission, yet some of the initial criticism has not been fully addressed:

(1) In the revised Fig.1C, it is still not evident to evaluate the conclusions from the authors. I am slightly surprised that the GFP signal dilutes, yet mitochondria seem maintain their exact shape and location in every single cell during the 60 minutes of the assay, which suggests that no fusion/fission events really occurred. Therefore, how can then be the lower GFP signal attributed to mitochondrial dynamics events?

(2) The shocking decrease in mtDNA remains unexplained (problem with replication?). Is it expected that this leads to relatively small changes in mitochondrial encoded genes.

(3) The TMEM135 imaging experiments remain unconvincing. The authors provide evidence that the antibody is specific, yet the signal does not seem to be purely mitochondrial and, for some reasons, all images are dramatically saturated. Lower saturation would be desirable. In the GFPAT KD group it seems as if TMEM135 translocates to the nucleus, which should be discussed further.

(4) TMEM135 knockdown impairs maximal respiration, yet not routine respiration in these cells. Still, the effect could be a higher sensitivity to the toxic effects of uncoupling agents, rather than a problem with respiration necessarily. In case lower uncoupling concentrations cannot be tested, the authors could use isolated mitochondria to solidify this point.

(5) Fig.6A continues to be a bit difficult to interpret. The authors argue that the Tomm20 antibody leads to a punctuated staining, and I have no doubt that they did their best to optimise their images. But wouldn't it be better to look for alternative markers to visualise mitochondrial architecture in a more faithful way? If the staining with Tomm20 does not allow for a good visualisation of the mitochondrial network, the authors might fail to appreciate the real overlap coefficient between mitochondria and peroxisomes.

Minor:

- Extended data Fig.2F: is it expected that b-actin is so evenly distributed across cellular compartments?

- In relation to the experiments in Extended Data 5H, my comment on its preliminary nature was not related to the technical proficiency of the experiments - apologies to the authors if my wording was led them to think this way. It is a very interesting experimental setup and result, but making a solid conclusion on metabolic flexibility based solely on ambient CO₂ measurements might be a stretch, specially when differences in RER dynamics were not observed at room temperature.

- Figure ED5I: I understand that the data was normalised by gram of tissue. However, the fact that lipid content in BAT is very different (see Fig.5H) means that the difference might simply be related to the different "lean" mass in the WT and TG brown adipose tissues. This should at least be acknowledged or discussed.

Reviewer #2 (Remarks to the Author):

This Reviewer wishes to salute the authors for seriously considering my issues with their initial submission and addressing all my concerns. In particular, the revised fluorescence microscopy images greatly reinforced the conclusions drawn by the authors.

Reviewer #3 (Remarks to the Author):

The Authors have performed critical additional work to improve the manuscript. In particular, this included using a different imaging approach to provide better and more convincing micrographs throughout the manuscript, with associated quantification, as well as including subfractionation data. A few additional minor points:

1. The graph in 1C has no y-axis label.
2. For 1C – the figure legend should also include the number of cells and number of experiments that data in the graph in C represent. In general, I think it is important to include both the number of experiments and the number of cells per experiment for all microscopy quantification.
3. The images in 1C are still a little confusing. As I understand the expt, at time 0, a proportion of the image is photoactivated to provide a GFP signal in this region. In the control cells, this appears as a quite a widespread yellow signal (GFP + TMRE). In contrast, in the PEX16 KO cells, there is a very limited green signal (proposed to be in mito that have photobleached TMRE), but almost not discernible yellow signal. I would expect the images at time = 0 to be the same for the control and KO lines. Is this incorrect? Since the quantitation is provided, this isn't a major point, but just to highlight that these images are confusing.
4. In their Response to Reviewers, the Authors mention that they have repeated some of the analysis blinded. I encourage them to include a statement about this in the manuscript (and the data in supp) as this provides a great deal of confidence in the data presented.
5. For immunoprecipitation, was the lysate centrifuged to pellet unlysed material before the IP? The reason I ask is that, with membrane proteins, it is possible to observe an "interaction" by IP if there is insufficient lysis of membranes. This is because you end up precipitating proteins within the same membrane. Please include this in the method if a centrifugation step was used.

RE: “TMEM135 Links Peroxisomes to the Regulation Brown Fat Mitochondrial Fission and Energy Homeostasis” by Hu et al.

Manuscript No.: NCOMMS-22-43318A

We thank the Editors and Reviewers for their valuable comments and the opportunity to improve our manuscript. We have revised the manuscript to incorporate reviewers' suggestions and editorial requests. Major changes in the manuscript are highlighted using red text color. Our point-by-point responses to reviewers' comments are provided below.

Reviewer #1 (Remarks to the Author):

The revised version of the manuscript shows some clear improvements compared to the original submission, yet some of the initial criticism has not been fully addressed:

Response: We appreciate your recognition of the improvements made based on your initial comments. We have addressed your remaining concerns to further enhance the quality of our manuscript.

(1) In the revised Fig.1C, it is still not evident to evaluate the conclusions from the authors. I am slightly surprised that the GFP signal dilutes, yet mitochondria seem maintain their exact shape and location in every single cell during the 60 minutes of the assay, which suggests that no fusion/fission events really occurred. Therefore, how can then be the lower GFP signal attributed to mitochondrial dynamics events?

Response: Thank you for your comments. We agree that it was difficult to assess mitochondrial dynamics in the images we presented previously. We repeated the assay using PA-GFP in Pex16 KO and control cells in order to improve image acquisition and now report zoomed in images to facilitate evaluation of mitochondrial fission and fusion events and quantify changes in the amount/intensity of fluorescence. Our new results with improved image quality presented in **revised Fig. 1C** suggest that mitochondria in control adipocytes are more actively involved in fission and fusion events as compared to those in Pex16 KO cells.

(2) The shocking decrease in mtDNA remains unexplained (problem with replication?). Is it expected that this leads to relatively small changes in mitochondrial encoded genes.

Response: The decreased mtDNA is consistent with the notion that mitochondrial dynamics profoundly impact mtDNA homeostasis through regulation of mtDNA replication and integrity. We added the following brief text on page 5 of the revised manuscript to address the reviewer's comments, mention a limitation of this assay (i.e. it is based on quantification of a single mtDNA-encoded gene) and to note that our results using this assay are very similar to those obtained by others in Drp1 knockdown cells:

“Replication of mtDNA is coordinated with fission events. Contacts between ER and mitochondria, which mediate mitochondrial fission, are necessary to permit mtDNA replication, though the underlying molecular mechanisms remain to be defined²⁶. In addition, disruption of mitochondrial dynamics affects the integrity of mtDNA. For example, Opa1 mutants exhibit mtDNA instability²⁷⁻²⁹. Pex16 KO resulted in a marked decrease in mtDNA content, which was measured using a PCR-based assay to assess relative

levels of the 12S mitochondrial rRNA gene (**Fig.1d**). Using this assay, similar results were observed in Drp1 knockdown cells³⁰.”

We have also revised the y-axis label for all mtDNA assay figure panels to specify that the assay reflects measurement of the relative levels of 12S mitochondrial rRNA gene. In addition, we repeated measurement of mtDNA content in TMEM135 KD in brown adipocytes to validate these results. As shown in **Figure A** below, our results are consistent with those reported in the manuscript (**Fig. 3f**).

Gene expression analysis indicated that certain mtDNA-encoded genes, such as MtCO1 and MtND6, were significantly decreased, while other mitochondrial transcripts were only minimally decreased (**Extended Data Fig. 4b**). As noted above, this likely reflects the multifactorial regulation of mtDNA by mitochondrial dynamics, including through effects on mtDNA stability.

Figure A mtDNA copy number normalized to nuclear DNA in SC and TMEM135 KD brown adipocytes measured by qPCR; n=5.

(3) The TMEM135 imaging experiments remain unconvincing. The authors provide evidence that the antibody is specific, yet the signal does not seem to be purely mitochondrial and, for some reasons, all images are dramatically saturated. Lower saturation would be desirable. In the GFPAT KD group it seems as if TMEM135 translocates to the nucleus, which should be discussed further.

Response: We noted in the manuscript that TMEM135 is present in multiple subcellular compartments. In Extended Data Fig. 2, subcellular fractionation in brown adipocytes indicates that TMEM135 is present in peroxisomes, mitochondria, and the nucleus. Therefore, the TMEM135 signal is not purely mitochondrial. We have reduced the saturation in **Fig. 2f**, **Fig. 2h** and **Extended Data Fig. S6a**.

Nuclear localization of TMEM135 is not a focus of the current manuscript, but is briefly discussed on page 16 of the revised manuscript:

“Surprisingly, disruption plasmalogen synthesis results in nuclear translocation of TMEM135. Physiological significance of the nuclear localization is unknown but might reflect a potential role for the protein in mitochondria-to-nucleus signaling. Additional work is needed to investigate this possibility.”

(4) TMEM135 knockdown impairs maximal respiration, yet not routine respiration in these cells. Still, the effect could be a higher sensitivity to the toxic effects of uncoupling agents, rather than a problem with respiration necessarily. In case lower uncoupling concentrations cannot be tested, the authors could use isolated mitochondria to solidify this point.

Response: Although basal respiration trended lower in TMEM135 KD mouse (**Fig. 3h**) or human (**Fig. 7e**) brown adipocytes, significant differences were only observed during FCCP-treated/maximal respiration. Since standard concentrations of Seahorse reagents, including FCCP, were used, it is unlikely that the decreased maximal respiration is due to a toxic effect of uncoupling reagents. Nevertheless, we isolated mitochondria from SC and TMEM135 KD brown adipocytes and measured complexes II- and IV-dependent respiration to avoid using uncoupling reagents. As shown in **Figure B** below, TMEM135 KD dramatically impaired mitochondrial respiration. Since we show in the manuscript that TMEM135 inactivation decreases mitochondrial respiration in cultured mouse and human adipocytes (**Fig. 3h** and **Fig. 7e**) and in frozen BAT samples from mice (**Extended Data Fig. 4e**), including the results shown below in the manuscript would be redundant.

Figure B OCR measured using a Seahorse XF24 Extracellular Flux Analyzer in mitochondria isolated from SC and TMEM135 KD brown adipocytes. Succ+R, Succinate + Rotenone; R+A, Rotenone + antimycin A; TMPD+ASC, N,N,N',N'-tetramethyl-p-phenylenediamine + Ascorbic acid; n = 10.

(5) Fig.6A continues to be a bit difficult to interpret. The authors argue that the Tomm20 antibody leads to a punctuated staining, and I have no doubt that they did their best to optimise their images. But wouldn't it be better to look for alternative markers to visualise mitochondrial architecture in a more faithful way? If the staining with Tomm20 does not allow for a good visualisation of the mitochondrial network, the authors might fail to appreciate the real overlap coefficient between mitochondria and peroxisomes.

Response: Although it is readily possible to show mitochondria morphology in cultured brown adipocytes (by immunofluorescence or using TMRE or mito-GFP, as we have shown in the manuscript), to our knowledge no one has reported mitochondrial morphology using immunofluorescence analysis in mouse frozen BAT tissue sections. That was why we decided to use TEM to assess mitochondria shape in mouse BAT (**Fig. 4d**). We tried different antibodies against Tomm20 (Proteintech #11802-1-AP) and another mitochondrial marker Tomm70 (CST#65619). Unfortunately, these antibodies also led to punctate staining of mitochondria in frozen BAT sections (see **Figure C**). Similar to our results, others have reported that immunostaining in BAT histology sections using Tomm20 antibodies shows punctate staining (PMID: 24713652 and PMID: 33956354). Although we believe that the punctate staining represents mitochondria, the inability to visualize full mitochondrial architecture using immunofluorescence might reflect partial masking of antigen in frozen tissue sections. Thus, we have decided to remove the Tomm20 staining data using BAT (**Fig. 6a**), especially given that we show peroxisome-mitochondria interaction by multiple methods. This includes 1) confocal microscopy showing that beta-adrenergic stimulation in cultured brown adipocytes promotes interaction between peroxisomes and mitochondria (with mitochondrial architecture clearly depicted) (originally **Extended Data Fig. 7a**; now **Fig. 6a**) and 2) a biochemical approach showing that TMEM135 translocates from peroxisomes to mitochondria in response to NE treatment in brown adipocytes (**Fig. 6c**).

Figure C. Immunofluorescence analysis using antibodies against PMP70 together with Tomm20 (upper panel) or Tomm70 (lower panel) in BAT sections from mice kept at room temperature.

- Extended data Fig.2F: is it expected that b-actin is so evenly distributed across cellular compartments?

Response: Although b-actin is ubiquitously present, short exposure Western blot (**Figure D**) shows that it is not quite evenly distributed across different subcellular compartments.

Figure D Western blot analysis of TMEM135 and organelle markers in whole cell lysates (WCL) and various subcellular fractions of differentiated brown pre-adipocytes from the wild-type C57BL/6J mice.

- In relation to the experiments in Extended Data 5H, my comment on its preliminary nature was not related to the technical proficiency of the experiments - apologies to the authors if my wording was led them to think this way. It is a very interesting experimental setup and result, but making a solid conclusion on metabolic flexibility based solely on ambient CO₂ measurements might be a stretch, specially when differences in RER dynamics were not observed at room temperature.

Response: We appreciate the reviewer's point. Thus, we have deleted the following statement on page 11 of the manuscript with regard to **Extended Data Fig. 5h**: "This suggests that TMEM135 overexpression interferes with the ability of mice to adapt to changing energy demands induced by thermoneutrality."

- Figure ED5I: I understand that the data was normalised by gram of tissue. However, the fact that lipid content in BAT is very different (see Fig.5H) means that the difference might simply be related to the different "lean" mass in the WT and TG brown adipose tissues. This should at least be acknowledged or discussed.

Response: Thank you for pointing this out. We agree with the reviewer that the lipid content in BAT is very different between WT and TMEM135^{TG} mice with HFD challenge. However, we note that a "lean" BAT generally reflects increased thermogenic activity. To recognize this point, we added the following statement on page 11:

"The increased glucose uptake in transgenic BAT is consistent with a more thermogenically active brown fat with markedly decreased intracellular lipid content (see **Fig. 5h**)."

Reviewer #2 (Remarks to the Author):

This Reviewer wishes to salute the authors for seriously considering my issues with their initial submission and addressing all my concerns. In particular, the revised fluorescence microscopy images greatly reinforced the conclusions drawn by the authors.

Response: We thank the Reviewer for the valuable comments and the opportunity to improve our manuscript.

Reviewer #3 (Remarks to the Author):

The Authors have performed critical additional work to improve the manuscript. In particular, this included using a different imaging approach to provide better and more convincing micrographs throughout the manuscript, with associated quantification, as well as including subfractionation data.

Response: We thank the reviewer for the helpful comments and suggestions that have improved our manuscript.

A few additional minor points:

1. The graph in 1C has no y-axis label.

Response: The y-axis label has been added in the **Fig. 1C**.

2. For 1C – the figure legend should also include the number of cells and number of experiments that data in the graph in C represent. In general, I think it is important to include both the number of experiments and the number of cells per experiment for all microscopy quantification.

Response: Thank you for this suggestion. The number of experiments and number of cells per experiment (when DAPI was used) is now indicated in figure legends. In addition, we added the following statement on page 24:

“In imaging experiments, the number of experiments is indicated in figure legends and if the cells were counter-stained with DAPI, the number of cells used in quantification is also indicated.”

3. The images in 1C are still a little confusing. As I understand the expt, at time 0, a proportion of the image is photoactivated to provide a GFP signal in this region. In the control cells, this appears as a quite a widespread yellow signal (GFP + TMRE). In contrast, in the PEX16 KO cells, there is a very limited green signal (proposed to be in mito that have photobleached TMRE), but almost not discernible yellow signal. I would expect the images at time = 0 to be the same for the control and KO lines. Is this incorrect? Since

the quantitation is provided, this isn't a major point, but just to highlight that these images are confusing.

Response: We appreciate this point. Although equal size area was photoactivated at T=0 for Pex16 KO and control adipocytes, it appears that the distribution of mitochondria in the photoactivated regions was not equal. As discussed in response to Reviewer #1 (1), we provided new images that address your concerns.

4. In their Response to Reviewers, the Authors mention that they have repeated some of the analysis blinded. I encourage them to include a statement about this in the manuscript (and the data in supp) as this provides a great deal of confidence in the data presented.

Response: Thank you for this suggestion. We added the following statement on page 24: "In selected imaging experiments (when indicated in figure legends), quantification was performed with the investigator blinded to the identity of the samples."

5. For immunoprecipitation, was the lysate centrifuged to pellet unlysed material before the IP? The reason I ask is that, with membrane proteins, it is possible to observe an "interaction" by IP if there is insufficient lysis of membranes. This is because you end up precipitating proteins within the same membrane. Please include this in the method if a centrifugation step was used.

Response: Yes, cell lysates were centrifuged to remove unlysed material prior to the IP. The method has been edited to include the following statement on page 20:

"Cell lysates were centrifuged at 13,000 RPM in a microcentrifuge for 10 minutes to remove unlysed cells. Supernatants were collected and subjected to protein quantification using BCA assay."

REVIEWERS' COMMENTS

Reviewer #1 (Remarks to the Author):

This referee wants to thank the authors for their efforts to address the comments raised during the revision process. This revised version is major improvement from their original submission and makes a strong case for the role of TMEM135 in mitochondrial fission and thermogenesis regulation in brown adipose tissue.

RE: “TMEM135 Links Peroxisomes to the Regulation Brown Fat Mitochondrial Fission and Energy Homeostasis” by Hu et al.

Manuscript No.: NCOMMS-22-43318B

Reviewer #1 (Remarks to the Author):

This referee wants to thank the authors for their efforts to address the comments raised during the revision process. This revised version is major improvement from their original submission and makes a strong case for the role of TMEM135 in mitochondrial fission and thermogenesis regulation in brown adipose tissue.

Response:

We thank the Reviewer for providing highly constructive and insightful comments during the entire revision process that greatly improved our paper.